# UniViT: Unifying Image and Video Understanding in One Vision Encoder

**Feilong Tang**[1,3,*] **Xiang An**[2*,†] **Haolin Yang**[3*]**, Yin Xie**[2]**, Kaicheng Yang**[2]**,**
**Ming Hu**[1,3]**, Zheng Cheng**[2]**, Xingyu Zhou**[2]**, Zimin Ran**[4]**, Imran Razzak**[3]**,**
**Ziyong Feng**[2]**, Behzad Bozorgtabar**[5]**, Jiankang Deng**[6]**, Zongyuan Ge**[1]
[1]Monash University, [2]DeepGlint, [3]MBZUAI, [4]UTS, [5]EPFL, [6]Imperial College London
`Feilong.Tang@monash.edu,xiangan@deepglint.com`

## Abstract

Despite the impressive progress of recent pretraining methods on multimodal tasks, existing methods are inherently biased towards either spatial modeling (*e.g.*, CLIP) or temporal modeling (*e.g.*, V-JEPA), limiting their joint capture of spatial details and temporal dynamics. To this end, we propose **UniViT**, a cluster-driven unified self-supervised learning framework that effectively captures the structured semantics of both image spatial content and video temporal dynamics through event-level and object-level clustering and discrimination. Specifically, we leverage offline clustering to generate semantic clusters across both modalities. For videos, multi-granularity event-level clustering progressively expands from single-event to structured multi-event segments, capturing coarse-to-fine temporal semantics; for images, object-level clustering captures fine-grained spatial semantics. However, while global clustering provides semantically consistent clusters, it lacks modeling of structured semantic relations (*e.g.,* temporal event structures). To address this, we introduce a contrastive objective that leverages these semantic clusters as pseudo-label supervision to explicitly enforce structural constraints, including temporal event relations and spatial object co-occurrences, capturing structured semantics beyond categories. Meanwhile, UniViT jointly embeds structured object-level and event-level semantics into a unified representation space. Furthermore, UniViT introduces two key components: *(i)* Unified Rotary Position Embedding integrates relative positional embedding with frequency-aware dimension allocation to support position-invariant semantic learning and enhance the stability of structured semantics in the discrimination stage; and *(ii)* Variable Spatiotemporal Streams adapt to inputs of varying frame lengths, addressing the rigidity of conventional fixed-input approaches. Extensive experiments across varying model scales demonstrate that UniViT achieves state-of-the-art performance on linear probing, attentive probing, question answering, and spatial understanding tasks.

## 1 Introduction

Visual representations are fundamental to the success of various downstream tasks. Contrastive [48, 71] and self-supervised [6, 10, 44] frameworks have strong spatial semantic modeling and cross-modal alignment. However, existing methods exhibit inherent biases toward specific modalities, limiting their joint capture of spatial details and temporal dynamics: Image-centric models (*e.g.,* CLIP [48]) capture static semantics but inadequately model temporal dynamics, while video-centric models (*e.g.,* V-JEPA [10]) incorporate temporal cues but exhibit deficiencies in fine-grained spatial modeling.

---

*Equal Contribution.
†Project lead.

39th Conference on Neural Information Processing Systems (NeurIPS 2025).

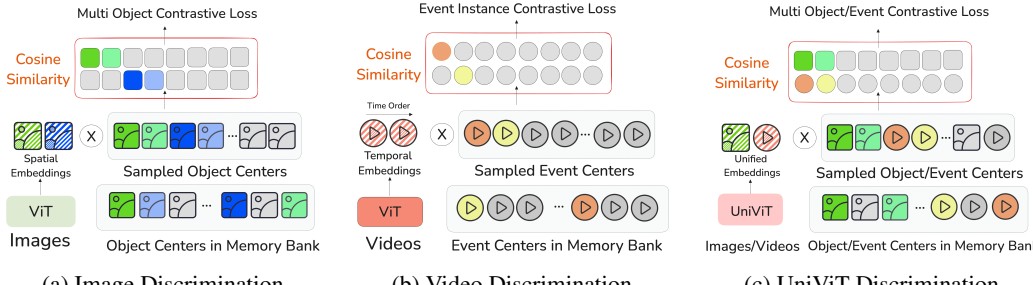

| (a) Image Discrimination | (b) Video Discrimination | (c) UniViT Discrimination |

Figure 1: Comparisons of cluster discrimination in image, video, and unified representation. (a) Image multi-label cluster discrimination improves semantic cohesion by assigning multiple samples to cluster centers, capturing various granularities of visual signals at the object level, but is limited to image representations. (b) Video cluster discrimination assigns discrete event-level labels to video segments, modeling dynamic semantics but lacking fine-grained spatial structures. (c) The proposed UniViT adopts unified multi-label discrimination at event and object levels with shared clusters and encoders, bridging spatial semantics and temporal structures within a unified representation.

The key to unified visual pretraining lies in jointly modeling static and dynamic semantics through structured, semantically coherent representations. Recent approaches such as UNICOM [2], MLCD [4] and RICE [64] enhance perception of structured semantics in images by introducing clustering mechanisms. MLCD further advances this approach by employing multi-label clustering to capture multiple semantic components in images, as depicted in Fig. 1 (a). Furthermore, Chat-Univi [34] introduces video event clustering for semantic modeling; however, it remains primarily focused on static objects and isolated events, lacking semantic modeling at the structured event level, thus failing to capture structured temporal dynamics. For instance, although actions such as "grabbing a cup" and "grabbing a phone" differ in their visual manifestation, a model with event-level abstraction can generalize them into a unified "grabbing" event category and capture the semantic relationship between the action and the target object. Moreover, such a model can infer the structural role of an action within an event sequence, such as recognizing that "grabbing a bowl" often precedes the event of "serving food." This capability signifies a transition from isolated event recognition to structured event understanding. Therefore, *we argue that transitioning from instance semantic recognition to cross-modal structured modeling is essential for constructing a unified visual pretraining framework.*

In this work, we propose UniViT, a cluster-driven unified self-supervised learning framework that effectively captures the structured semantics of both image spatial content and video temporal dynamics through event-level and object-level clustering and discrimination, as depicted in Fig. 1(c). Specifically, we design a two-stage cluster-discrimination training paradigm. In the clustering stage, we employ offline clustering to generate semantic for both modalities. For videos, we perform multi-granularity event-level clustering by densely sampling frames at multiple temporal scales, progressively organizing individual events into structured segments of multiple events, thus capturing coarse-to-fine temporal semantics. For static images and individual video frames, we employ object-level clustering extracts fine-grained spatial semantics. Subsequently, in the discrimination stage, we introduce a contrastive objective that utilizes these semantics as pseudo-label supervision to explicitly enforce structural constraints, including temporal relations among adjacent events and spatial co-occurrences among objects. This approach captures structured semantics beyond isolated categories, aligning dynamic and static semantic content within a unified representation space.

The core of this method is to jointly model structured semantics across image and video modalities within a unified representation space, effectively bridging static spatial details and dynamic temporal relations. Therefore, UniViT introduces two critical components: *(i)* Unified Rotary Position Embedding (U-RoPE), decomposing positional embeddings into distinct spatial and temporal components through frequency-aware allocation, thereby facilitating position-invariant representation learning and enhancing stability of structured semantic representations during the discrimination stage; and *(ii)* Variable Spatiotemporal Streams (VS$^2$), adapting to varying frame lengths, enabling the model to flexibly capture fine-grained spatial details and diverse temporal scales simultaneously. Notably, UniViT retains the original vision encoder without incurring extra inference costs.

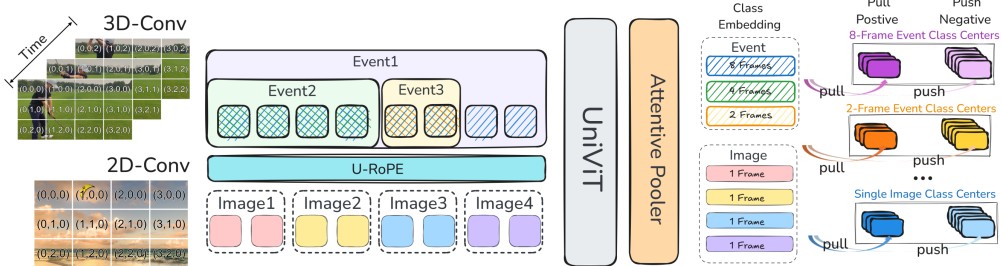

Figure 2: **Overview of the UniViT framework for unified representation learning.** Image and video inputs are processed by 2D/3D convolutions, with video features segmented into events. The resulting tokens are fed into a shared UniViT backbone with U-RoPE, followed by Attentive Pooling to produce a class-level representation for contrastive learning against memory bank class centers.

Extensive experiments across varying model scales demonstrate that UniViT achieves state-of-the-art performance on multiple downstream tasks, including linear probing, attentive probing, image and video question answering, and spatial understanding. The contributions are summarized as follows: *(i):* We propose UniViT, a cluster-driven unified self-supervised learning framework that effectively captures the structured semantics of both image spatial content and video temporal dynamics through event-level and object-level clustering and discrimination. *(ii):* We introduce U-RoPE and $VS^2$ strategies to explicitly disentangle spatial-temporal positional embeddings and adaptively handle varying frame lengths, facilitating position-invariant and structured spatiotemporal representations. *(iii):* Extensive experiments across diverse model scales and tasks demonstrate that UniViT achieves state-of-the-art performance on multiple downstream tasks.

## 2 Methodology

Our goal is to achieve unified representation learning across image and video modalities (Fig. 2).

**Variable Spatiotemporal Streams ($VS^2$).** Given visual samples $X = \{x_i\}_{i=1}^N$, each sample $x_i$ is uniformly divided into non-overlapping patches, accommodating both images and videos in a unified manner. Specifically, each image sample $x_i \in \mathbb{R}^{H \times W \times 3}$ is partitioned into spatial patches of size $P \times P$, while each video sample $x_i \in \mathbb{R}^{T \times H \times W \times 3}$ is partitioned into variable-length frame sequences $\{\mathcal{F}_{i,s}\}_{s=1}^{S_i}$, with each sequence $\mathcal{F}_{i,s} \in \mathbb{R}^{T_{i,s} \times H \times W \times 3}$ covering consecutive RGB frames (*e.g.*, 1, 2, 4, 8, or 16 frames) over the same spatial regions, where $S_i$ denotes the number of sequences within the $i$-th video. Subsequently, each sequence $\mathcal{F}_{i,s}$ is independently encoded via a shared Transformer-based encoder $\phi(\cdot)$ into spatiotemporal embedding tokens $Z_i$:

$$Z_i = [\,\phi(\mathcal{F}_{i,1}); \, \phi(\mathcal{F}_{i,2}); \, \ldots; \, \phi(\mathcal{F}_{i,S_i})\,] \in \mathbb{R}^{M \times C}, \quad \text{with} \quad M = \sum_{s=1}^{S_i} M_s. \tag{1}$$

where $M_s$ and $C$ denote the number of embedding tokens and the embedding dimension for the $s$-th sequence, respectively. The resulting tokens are projected into $D$-dimensional vectors, forming token embeddings $E_i = \{e_{i,j}\}_{j=1}^M \in \mathbb{R}^{M \times D}$ that encode local visual features [21]. Subsequently, positional encodings are dynamically assigned using the $VS^2$ strategy. Therefore, this design enhances flexibility and spatiotemporal representational capacity.

**Event-level and Object-level Clustering.** Iterative clustering-discrimination approaches commonly suffer from substantial computational overhead [14]. To address this issue, we adopt a single-step offline clustering to efficiently capture both object-level semantics from images and event-level semantics from videos. Specifically, image embeddings are obtained by pooling the features extracted from local object patches, $e_i^{obj} = e_i \in \mathbb{R}^D$, while video embeddings which are derived from a fixed-length 16-frame input are obtained by concatenating frame-level features within each segment, yielding $e_{i,s}^{\text{evt}} = [e_{i,1}; \ldots; e_{i,s}] \in \mathbb{R}^{s \times D}$ from variable-length sub-clips of $s \in \{1; 2; \ldots; S_i\}$. We define a set of shared semantic centroids $\mathcal{C} = \{c_k^{\text{obj}}\}_{k=1}^{K_{\text{obj}}} \cup \{c_k^{\text{evt}}\}_{k=1}^{K_{\text{evt}}} \subseteq \mathbb{R}^D$, where $K = K_{\text{obj}} + K_{\text{evt}}$ represents the total number of clusters across both modalities. The clustering objective is then

formulated separately for object and event embeddings:

$$\mathcal{C}_{\text{uni}} = \arg \min_{\{c_k^{\text{obj}}, c_k^{\text{evt}}\}} \sum_{i=1}^{N} \left( \min_{k \in [1, K_{\text{obj}}]} \|e_i^{\text{obj}} - c_k^{\text{obj}}\|_2^2 + \sum_{s=1}^{S_i} \min_{k \in [1, K_{\text{evt}}]} \|e_{i,s}^{\text{evt}} - c_k^{\text{evt}}\|_2^2 \right), \quad (2)$$

where $N$ is the number of samples. $\mathcal{C}_{\text{uni}}$ integrates object-level and event-level semantics for consistent representation learning.

**Unified Rotary Position Embedding (U-RoPE).** Unlike traditional absolute position encoding defined as $p = (t, x, y)$, U-RoPE adopts a relative scheme [53] $\Delta p = (t_1 - t_2, x_1 - x_2, y_1 - y_2)$ that supports position-invariant semantic learning, enabling better modeling of multi-event structures in videos. Specifically, the rotary position embedding is applied directly to the query-key dot-product attention matrix, *i.e.,* $\mathbf{A}_{i,j} = (\mathbf{q}_i R_i)(\mathbf{k}_j R_j)^{\top}$. For image inputs, the temporal position $t$ is fixed across the spatial grid $(x, y)$. For video inputs, temporal positions vary across frames while spatial positions are computed the same way as for images. Existing methods, such as M-RoPE [60], typically allocate temporal position encodings with high-frequency components, determined by the rotary frequency $\theta_n = \beta^{-\frac{2n}{C}}$. This allocation causes periodic oscillations, leading to unstable frame representations that conflict with dense label discrimination. Therefore, we propose a unified frequency allocation strategy that assigns global event-related temporal structures to smoother low-frequency components, while retaining high-frequency components for local spatial details:

$$\Phi_S = \{\beta^{-\frac{2(2j+k)}{C}} \mid j \in [0, \frac{3}{4}L), k \in \{0, 1\}\}, \quad \Phi_T = \{\beta^{-\frac{2j}{C}} \mid j \in [\frac{3}{4}L, L)\}, \quad (3)$$

where $\Phi_S$ and $\Phi_T$ respectively denote the rotation frequencies used for spatial and temporal rotary applying to the $2L$-dimensional embedding space, with $L = \frac{C}{2}$, and $\beta$ represents the base frequency.

**Joint Training Objective.** Visual samples commonly exhibit multiple semantic components, including object-level semantics from images and event-level semantics from videos, rendering single-label assignments inadequate for unified multimodal representation learning. To capture both object-level and event-level semantic structures, we introduce a contrastive objective that leverages these semantic clusters as pseudo-label supervision to explicitly enforce structural constraints. Specifically, for each visual embedding $e_i \in \mathbb{R}^D$, we identify multiple positive semantic labels from the unified semantic centroid set $\mathcal{C}_{uni} \in \mathbb{R}^{(|\mathcal{C}_{obj}| + |\mathcal{C}_{evt}|) \times D}$, consisting of both object-level $\mathcal{C}_{obj}$ and event-level $\mathcal{C}_{evt}$ centroids. The remaining centroids in this unified set are treated as negative labels. Subsequently, the joint multi-label semantic discrimination objective is formulated as:

$$\mathcal{L}_{Joint} = \sum_{m \in \{obj, evt\}} \left[ \log(1 + \sum_{j \in \Omega_n^m} \exp(\sigma_j^m)) + \log(1 + \sum_{i \in \Omega_p^m} \exp(-\sigma_i^m)) \right], \quad (4)$$

where $m \in \{obj, evt\}$ denotes the semantic granularity level, corresponding respectively to object-level (images or single frames) and event-level (video segments). $\Omega_p^m$ and $\Omega_n^m$ represent the sets of positive and negative semantic labels for granularity level $m$, while $\sigma_i^m$ and $\sigma_j^m$ indicate embedding similarity scores to positive and negative semantic centroids, respectively. The embedding similarity score $\sigma_{u,k}^m$ is computed as $\sigma_{u,k}^m = e_u^{\top} c_k^m$, where $u$ and $k$ index visual embeddings and semantic centroids within the corresponding positive or negative sets, respectively. This unified formulation leverages semantic clustering to capture spatiotemporal structures for discriminative embeddings.

**Unified Image and Video Understanding.** As shown in Fig.3a, the frame-to-frame similarity under the multi-event setting is significantly lower than that of the single-event counterpart, indicating finer-grained temporal discrimination and better action segmentation. The lower training loss in Fig. 3b indicates that fine-grained temporal modeling in the multi-event setting benefits video understanding. In Fig. 3c, U-RoPE demonstrates faster convergence and improved stability over absolute position encoding and M-RoPE. By decoupling spatial and temporal dimensions and avoiding high-frequency temporal encoding, U-RoPE enables robust position-invariant learning across both images and videos. Fig. 3d illustrates the distribution of cosine similarity between image and video feature embeddings, indicating that video and image representations are related yet significantly different.

## 3 Experiments

### 3.1 Implementation Details.

**Pretraining Setup.** Our models are pre-trained on the LAION400M[50], COYO700M[13], and InternVid[61]. We use 80 H800 GPUs for the training process. During training, we maintained a 1:1

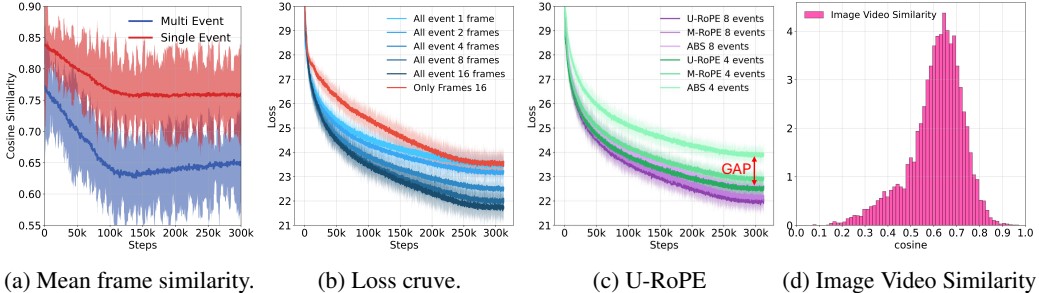

| (a) Mean frame similarity. | (b) Loss cruve. | (c) U-RoPE | (d) Image Video Similarity |

Figure 3: Analysis of multi-event approach. (a) Multi-event frames demonstrate significantly lower frame similarity (0.65 vs 0.75, indicating greater inter-frame distinctiveness that improves feature discrimination. (b) Multi-event processing substantially accelerates convergence speed, with higher frame counts (8-16) showing faster loss reduction compared to single or fewer frames. (c) U-RoPE enables better local semantic learning, resulting in lower loss compared to other methods. (d) Distribution of cosine similarity between image and video feature embeddings.

| Method | Arch. | Resolution | Params | Data Imgs/Videos | Language pre. | **Image Tasks** | | | | | | | | **Video Tasks** | | | | | | | |
|---|---|---|---|---|---|---|---|---|---|---|---|---|---|---|---|---|---|---|---|---|---|
| | | | | | | Avg. | Food [12] | CIFAR [36] | EuroSAT [32] | Resisc45 [18] | Caltech101 [62] | Imagenet [49] | Sun397 [63] | Avg. | K400 [35] | K600 [16] | K700 [51] | HMDB51 [37] | UCF101 [52] | RareAct [42] | SSV2 [27] |
| *Methods pretrained on Images* | | | | | | | | | | | | | | | | | | | | | |
| Siglipv2 [58] | ViT-B/16 | 256 | 86M | 10B | ✓ | 90.5 | 95.2 | 94.7 | 93.1 | 90.9 | 95.4 | 83.2 | 81.2 | 58.2 | 71.3 | 70.9 | 59.0 | 66.3 | 92.2 | 31.3 | 16.3 |
| | ViT-L/16 | 256 | 303M | 10B | ✓ | 92.6 | 96.5 | 97.2 | 94.5 | 94.3 | 96.0 | 84.1 | 85.3 | 63.9 | 77.3 | 77.5 | 66.5 | 71.7 | 95.0 | 39.8 | 19.4 |
| | ViT-SO400M | 224 | 428M | 10B | ✓ | **93.4** | **97.1** | **97.8** | 95.0 | **95.8** | **96.1** | 85.5 | 86.4 | 66.4 | 78.7 | 79.1 | 68.8 | 76.3 | 95.0 | 46.1 | 20.9 |
| DINOv2 [44] | ViT-S/14 | 224 | 22M | 145M | ✗ | 85.9 | 89.3 | 90.2 | 92.4 | 85.6 | 91.0 | 82.3 | 70.3 | 49.2 | 60.3 | 60.2 | 49.8 | 56.5 | 87.2 | 16.4 | 14.3 |
| | ViT-B/14 | 224 | 87M | 145M | ✗ | 89.3 | 95.1 | 94.7 | 92.1 | 90.4 | 93.5 | 83.1 | 76.4 | 56.7 | 66.9 | 66.9 | 54.3 | 63.8 | 91.6 | 38.8 | 14.5 |
| | ViT-L/14 | 224 | 311M | 145M | ✗ | 91.4 | 95.2 | 96.4 | 94.3 | 91.2 | 95.4 | 84.7 | 82.4 | 56.5 | 67.5 | 67.5 | 54.2 | 64.3 | 91.8 | 35.6 | 14.6 |
| CLIP [48] | ViT-B/16 | 224 | 86M | 400M | ✓ | 89.0 | 93.0 | 92.4 | 92.4 | 90.8 | 95.6 | 81.4 | 77.1 | 56.5 | 67.4 | 67.1 | 54.2 | 66.9 | 88.1 | 37.5 | 14.0 |
| | ViT-L/14 | 224 | 304M | 400M | ✓ | 91.6 | 95.7 | 94.2 | 94.5 | 93.4 | 95.7 | 83.2 | 84.7 | 63.7 | 74.8 | 74.2 | 63.2 | 71.3 | 92.6 | 53.1 | 16.8 |
| I-JEPA [7] | ViT-H/14 | 224 | 631M | 22K | ✗ | 83.0 | 76.3 | 94.7 | 89.7 | 79.5 | 92.3 | 80.1 | 68.4 | 40.6 | 48.6 | 46.6 | 34.1 | 56.3 | 83.9 | 2.0 | 12.4 |
| | ViT-g/16 | 224 | 1011M | 22K | ✗ | 85.0 | 78.9 | 95.1 | 90.4 | 83.7 | 93.7 | 83.3 | 70.2 | 41.5 | 48.9 | 47.2 | 35.3 | 58.2 | 83.1 | 3.9 | 13.8 |
| MLCD [4] | ViT-L/14 | 336 | 304M | 1.1B | ✗ | 92.9 | 96.8 | **97.8** | 95.0 | 95.2 | 95.7 | 84.3 | 85.4 | 66.8 | 78.8 | 78.9 | 68.2 | 80.4 | 97.1 | 46.9 | 17.4 |
| *Methods pretrained on Videos* | | | | | | | | | | | | | | | | | | | | | |
| V-JEPA [10] | ViT-L/16 | 224 | 312M | 2M | ✗ | - | - | - | - | - | - | - | - | 57.0 | 62.3 | 63.2 | 49.2 | 77.7 | 95.4 | 6.3 | **44.9** |
| | ViT-H/16 | 224 | 649M | 2M | ✗ | - | - | - | - | - | - | - | - | 56.1 | 59.3 | 60.5 | 46.9 | 83.0 | 94.4 | 4.0 | 44.8 |
| LanguageBind [73] | ViT-L/14 | 224 | 407M | 3M | ✓ | - | - | - | - | - | - | - | - | 60.0 | 72.1 | 72.3 | 60.3 | 71.9 | 93.3 | 33.8 | 16.4 |
| VideoMAEv2 [59] | ViT-g/14 | 224 | 1012M | 1.35M | ✗ | - | - | - | - | - | - | - | - | 35.3 | 39.8 | 42.6 | 29.4 | 30.6 | 75.0 | 1.6 | 27.9 |
| *Methods pretrained on Image and Videos* | | | | | | | | | | | | | | | | | | | | | |
| PE-Core [11] | ViT-B/16 | 224 | 93M | 5.4B/22M | ✓ | 89.9 | 94.0 | 92.7 | 92.9 | 91.2 | 94.4 | 84.2 | 80.1 | 54.0 | 66.6 | 66.2 | 53.6 | 65.9 | 90.3 | 19.4 | 15.9 |
| | ViT-L/16 | 336 | 317M | 5.4B/22M | ✓ | 92.1 | 96.7 | 94.5 | 94.8 | 93.6 | 95.3 | 85.3 | 84.6 | 67.6 | 79.6 | 79.8 | 69.4 | 77.7 | 96.8 | 47.5 | 22.5 |
| | ViT-G/14 | 448 | 1882M | 5.4B/22M | ✓ | 93.1 | 96.9 | 97.7 | **95.6** | 93.9 | 95.7 | **86.2** | 85.4 | 68.4 | 81.5 | 81.7 | 72.1 | 80.5 | 97.3 | 42.2 | 23.7 |
| UniViT | ViT-S/16 | 224 | 26M | 1.1B/60M | ✗ | 85.5 | 88.9 | 85.5 | 92.5 | 87.4 | 91.4 | 81.3 | 71.5 | 52.0 | 65.6 | 64.9 | 50.9 | 59.4 | 89.6 | 18.1 | 15.6 |
| | ViT-B/16 | 224 | 99M | 1.1B/60M | ✗ | 89.0 | 91.4 | 92.6 | 93.1 | 90.2 | 94.2 | 83.1 | 78.2 | 61.9 | 74.7 | 74.8 | 61.5 | 70.2 | 94.8 | 32.0 | 25.1 |
| | ViT-L/14 | 224 | 334M | 1.1B/60M | ✗ | 91.8 | 95.2 | 95.5 | 94.7 | 93.1 | 95.4 | 84.2 | 84.5 | 68.3 | 82.9 | 83.0 | 72.4 | 81.4 | 97.3 | 33.8 | 27.3 |
| | ViT-L/14 | 336 | 334M | 1.1B/60M | ✗ | 92.8 | 95.1 | 97.7 | 95.1 | 93.7 | 95.7 | 85.4 | **86.5** | **73.1** | **84.1** | **84.2** | **73.3** | **83.5** | **98.2** | **56.3** | 32.1 |

Table 1: *Attentive Probe Evaluation under few-shot settings for label efficiency analysis.* **Bold** indicates the best performance.

ratio between images and video frames, with an image batch size of 16K and a video batch size of 2K (each video containing 16 frames). In total, our model is exposed to approximately 20B image frames throughout the training. For our standard model, we use 224 resolution images. For the 336 resolution variant, we first train the model at 224 resolution, then increase it to 336 and continue training for an additional 1B frames. We utilize the AdamW optimizer with a learning rate of $0.001$ and weight decay of $0.2$. The number of classes ($k$) is one million, the ratio of sampled negative class centers ($r$) is 0.1, and the number of positive labels ($l$) assigned to each image and video is 8.

**Multimodal Setup.** For our multimodal large language model evaluations, we adopt the LLaVA-NeXT [41] framework while maintaining experimental consistency. All training methodologies precisely follow the original LLaVA-NeXT-Video implementation, utilizing identical pretraining datasets and instruction-tuning data. We employ Qwen2.5-7B [68] as our language model backbone, which effectively mitigates potential hyperparameter biases that might favor OpenAI-CLIP in the original LLaVA-NeXT-Video configuration. This controlled experimental design ensures fair comparison when evaluating the performance of our vision encoders within multimodal systems.

## 3.2  Comparisons with Existing Vision Encoders

**Attentive Probing Results.** We evaluate the comprehensive ability of UniViT across 14 standard benchmarks, covering a wide range of semantic and vision-centric tasks, using a 50-shot attentive

| Method | Arch. | Resolution | Params | Data Imgs/Videos | Language pre. | Image Tasks | | | | | | | | Video Tasks | | | | | | | |
|---|---|---|---|---|---|---|---|---|---|---|---|---|---|---|---|---|---|---|---|---|---|
| | | | | | | Avg. | Food [12] | CIFAR [36] | EuroSAT [32] | Resisc45 [18] | Caltech101 [62] | Imagenet [49] | Sun397 [63] | Avg. | K400 [35] | K600 [16] | K700 [51] | HMDB51 [37] | UCF101 [52] | RareAct [42] | SSV2 [27] |
| *Methods pretrained on Images* | | | | | | | | | | | | | | | | | | | | | |
| Siglipv2 [58] | ViT-B/16 | 256 | 86M | 10B | ✓ | 93.7 | 94.6 | 96.7 | 98.3 | 93.1 | 98.5 | 83.5 | 91.3 | 56.3 | 72.4 | 73.6 | 60.2 | 63.7 | 90.6 | 12.5 | 21.1 |
| | ViT-L/16 | 256 | 303M | 10B | ✓ | 94.3 | 96.2 | 97.8 | 98.2 | 93.1 | **98.6** | 85.2 | 91.3 | 59.4 | 76.5 | 78.2 | 66.2 | 66.8 | 93.1 | 12.5 | 22.7 |
| | ViT-SO400M | 224 | 428M | 10B | ✓ | 94.7 | 96.4 | 98.2 | 97.8 | 94.0 | **98.6** | 85.7 | **92.2** | 66.5 | 78.6 | 79.7 | 68.8 | 68.0 | 94.4 | 53.1 | 23.2 |
| DINOv2 [44] | ViT-S/14 | 224 | 22M | 145M | ✗ | 90.6 | 89.1 | 97.7 | 98.1 | 84.4 | 97.0 | 81.1 | 86.9 | 52.2 | 62.6 | 62.9 | 49.0 | 53.5 | 82.9 | 37.5 | 17.2 |
| | ViT-B/14 | 224 | 87M | 145M | ✗ | 92.1 | 92.8 | 98.7 | 98.1 | 86.1 | 96.1 | 84.5 | 88.1 | 59.0 | 68.9 | 70.6 | 57.3 | 61.7 | 89.9 | 46.9 | 18.0 |
| | ViT-L/14 | 224 | 311M | 145M | ✗ | 93.6 | 94.3 | **99.4** | 98.5 | 90.1 | 97.5 | 86.3 | 89.0 | 61.0 | 73.7 | 74.1 | 63.1 | 64.4 | 91.8 | 40.6 | 19.6 |
| CLIP [48] | ViT-B/16 | 224 | 86M | 400M | ✓ | 91.8 | 92.2 | 95.9 | 97.8 | 90.9 | 96.3 | 80.2 | 89.2 | 58.3 | 70.3 | 71.7 | 57.7 | 64.4 | 88.5 | 37.5 | 18.0 |
| | ViT-L/14 | 224 | 304M | 400M | ✓ | 93.8 | 95.0 | 98.1 | 98.6 | 93.3 | 97.4 | 83.9 | 90.5 | 64.4 | 76.4 | 77.8 | 65.8 | 64.8 | 92.8 | 53.1 | 19.8 |
| I-JEPA [7] | ViT-H/14 | 224 | 631M | 22K | ✗ | 86.1 | 74.1 | 98.3 | 98.6 | 78.6 | 95.6 | 79.3 | 78.0 | 41.8 | 49.4 | 50.2 | 37.5 | 46.9 | 74.1 | 18.8 | 15.5 |
| | ViT-g/16 | 224 | 1011M | 22K | ✗ | 87.9 | 77.7 | 98.2 | 98.7 | 82.5 | 95.8 | 82.1 | 80.6 | 42.0 | 49.6 | 50.4 | 37.7 | 47.7 | 74.7 | 18.8 | 15.1 |
| MLCD [4] | ViT-S/16 | 224 | 22M | 1.1B | ✗ | 88.7 | 84.0 | 94.2 | 98.5 | 87.9 | 92.9 | 79.1 | 84.6 | 38.9 | 48.1 | 49.6 | 38.3 | 36.7 | 61.4 | 25.0 | 13.0 |
| | ViT-B/16 | 224 | 86M | 1.1B | ✗ | 91.7 | 89.8 | 97.6 | 98.7 | 90.8 | 95.8 | 82.3 | 86.9 | 48.3 | 58.9 | 60.7 | 48.4 | 46.5 | 79.9 | 28.1 | 15.6 |
| | ViT-L/14 | 224 | 304M | 1.1B | ✗ | 91.4 | 87.2 | 97.2 | 98.8 | 89.3 | 95.6 | 85.4 | 86.0 | 58.1 | 71.4 | 72.8 | 60.0 | 60.2 | 88.3 | 37.5 | 16.7 |
| | ViT-L/14 | 336 | 304M | 1.1B | ✗ | 94.9 | 96.2 | **99.4** | **99.1** | 94.5 | 97.9 | 86.3 | 91.0 | 62.2 | 76.0 | 76.2 | 64.1 | 62.5 | 92.2 | 46.9 | 17.5 |
| *Methods pretrained on Videos* | | | | | | | | | | | | | | | | | | | | | |
| LanguageBind [73] | ViT-L/14 | 224 | 407M | 3M | ✓ | - | - | - | - | - | - | - | - | 63.6 | 74.4 | 75.1 | 62.5 | 71.1 | 92.9 | 46.9 | 22.1 |
| *Methods pretrained on Image and Videos* | | | | | | | | | | | | | | | | | | | | | |
| PE-Core [11] | ViT-B/16 | 224 | 93M | 5.4B/22M | ✓ | 93.4 | 93.2 | 98.1 | 98.8 | 92.9 | 97.2 | 83.4 | 90.5 | 58.1 | 69.5 | 71.2 | 56.8 | 66.0 | 88.7 | 37.5 | 16.9 |
| | ViT-L/14 | 336 | 317M | 5.4B/22M | ✓ | 95.0 | 96.2 | **99.4** | 98.8 | 93.6 | 98.0 | 86.7 | 92.0 | 63.7 | 74.5 | 75.4 | 62.2 | 69.1 | 92.4 | 53.1 | 19.0 |
| | ViT-G/14 | 448 | 1882M | 5.4B/22M | ✓ | **95.2** | 96.3 | 99.3 | 98.1 | 93.5 | 97.9 | **89.5** | 92.1 | 68.6 | 80.8 | 81.1 | 70.4 | 73.4 | 94.7 | 56.3 | 23.2 |
| UniViT | ViT-S/16 | 224 | 26M | 1.1B/60M | ✗ | 91.0 | 87.7 | 95.8 | 98.8 | 90.5 | 96.4 | 80.4 | 87.3 | 56.0 | 67.6 | 68.8 | 54.0 | 56.6 | 87.6 | 40.6 | 16.8 |
| | ViT-B/16 | 224 | 99M | 1.1B/60M | ✗ | 92.2 | 90.1 | 97.1 | 98.9 | 92.2 | 96.3 | 83.1 | 87.5 | 60.8 | 75.7 | 77.3 | 63.2 | 66.0 | 93.1 | 28.1 | 22.3 |
| | ViT-L/14 | 224 | 334M | 1.1B/60M | ✗ | 94.3 | 94.8 | 98.8 | 99.0 | 93.8 | 98.1 | 85.6 | 90.2 | 67.1 | 84.3 | 85.3 | 74.5 | 64.1 | 92.2 | 43.8 | 25.5 |
| | ViT-L/14 | 336 | 334M | 1.1B/60M | ✗ | 94.9 | **96.6** | 98.9 | 99.0 | **94.9** | 98.4 | 86.5 | 90.2 | **72.0** | **85.3** | **85.4** | **74.9** | **78.1** | **96.6** | **56.3** | 27.4 |

Table 2: *Linear Probe Evaluation.* **Bold** indicates the best performance.

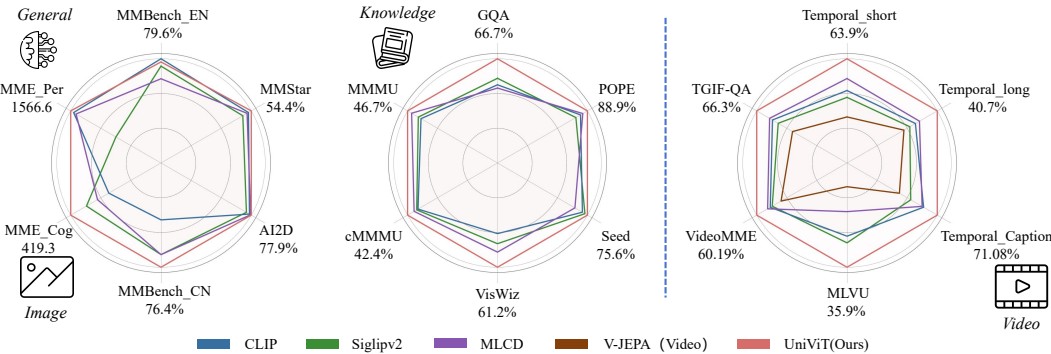

Figure 4: Comparisons of UniViT and existing vision encoders across diverse multimodal image and video benchmarks, highlighting UniViT's superior unified representation capabilities.

probing setup, which allows us to measure feature quality without extensive task-specific finetuning. As shown in Table 1, our model achieves state-of-the-art performance on multiple standard benchmarks, including object classification, scene recognition, and fine-grained categorization.

**Linear Probing Results.** As shown in Table 2, we report performance on the same set of benchmarks as in attentive probe, this time using a standard linear probing setup instead. The trend aligns closely with the attentive probe results; our UniViT achieves state-of-the-art performance on both semantic and non-semantic tasks, further validating the quality and generality of the learned representations.

All experimental settings for the compared models strictly follow their original implementations to ensure fair comparison. Under this consistent protocol, despite being pretrained without pixel-level or feature-level supervision (*e.g.,* MAE, JEPA), our UniViT still achieves strong visual understanding across a wide range of vision-centric tasks. Notably, previous video encoders cannot handle static images, while image encoders struggle with temporally dependent tasks (*e.g.,* SSV2). In contrast, UniViT leverages a unified representation space that captures both spatial and temporal patterns, enabling strong performance across image and video domains. This demonstrates the generalization capability of our architecture, which is designed for unified representation learning across modalities.

**UniViT as a Vision Encoder for MLLMs.** In this section, we evaluate our unified vision encoder, UniViT, which is designed to seamlessly handle both image and video modalities. We conduct comprehensive experiments on a diverse set of benchmarks to assess the model's ability to learn shared semantic representations across static and temporal inputs. The evaluation is performed within the LLaVA-NeXT-Video framework under consistent and controlled settings to ensure fair comparison. We benchmark UniViT on 18 datasets spanning four major domains: General VQA and Knowledge VQA. These datasets cover both image-based and video-based VQA tasks, providing

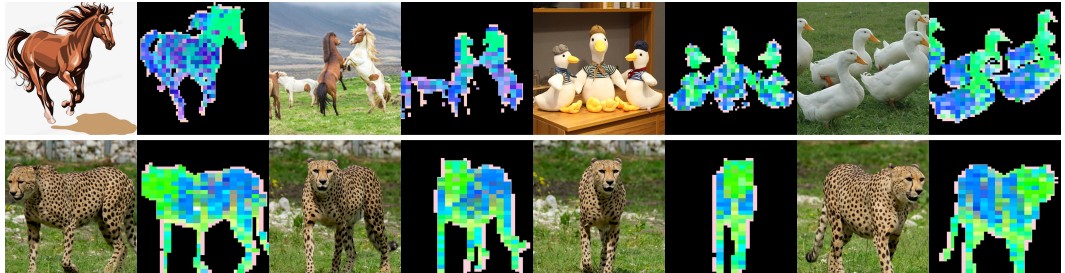

Figure 6: PCA visualization of patch features extracted by our UniViT. Patches displaying similar colors indicate semantic similarities, reflecting that they embody analogous elements or attributes.

a holistic view of the model's multimodal reasoning capability. As summarized in Fig. 4, UniViT achieves strong and consistent performance across all categories, notably outperforming conventional single-modality baselines such as CLIP and SigLIP. At a resolution of 336px, UniViT surpasses CLIP on 17 out of 18 benchmarks. Importantly, UniViT demonstrates robust performance across both Image-VQA and Video-VQA, highlighting its versatility and effectiveness in learning generalizable multimodal semantics. Notably, these results are achieved without any language supervision, further demonstrating the strength of our unified framework in capturing cross-modal visual understanding. This positions UniViT as a practical and scalable solution for real-world multimodal applications.

**Scaling Behavior.** To investigate the scalability of our unified vision encoder, we conduct a systematic analysis of its performance across varying configurations using an attentive probe protocol on both image and video tasks. Rather than comparing against other paradigms, we focus on the internal scaling behavior of our framework along three progressive dimensions: (1) increasing training data volume, (2) expanding model capacity, and (3) increasing input resolution.

As illustrated in Fig. 5, we analyze the scaling behavior of UniViT on video tasks by averaging performance scores from seven video datasets. This provides a stable estimate of performance trends specific to video understanding. This controlled design allows us to isolate and examine the contribution of each scaling factor. We observe consistent and meaningful improvements at all stages. Increasing data volume leads to noticeable gains for both modalities, suggesting that enhanced data diversity improves unified semantic modeling. Expanding model capacity, from small to large variants, further boosts representation quality, with more pronounced benefits on fine-grained tasks. Higher input resolution also contributes positively to overall performance, especially in video tasks where capturing spatial continuity across frames is crucial for robust representation learning. Compared to existing models, our approach exhibits more efficient scaling behavior, with larger variants

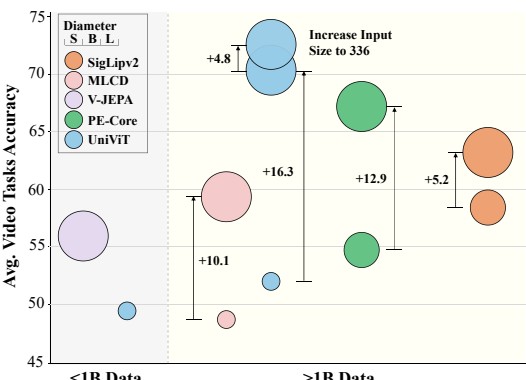

Figure 5: Scaling Behavior: Performance scalability of UniViT across model capacities, training data volumes, and input resolutions. Average accuracy across multiple video tasks demonstrates consistent improvements with increased scale.

delivering consistently stronger improvements across video tasks. These results demonstrate that our model scales effectively with standard training resources while maintaining strong generalization across diverse video benchmarks.

## 4 Ablation Study

**Pretraining Strategy.** As shown in Table 3a and 3b, we conduct ablation studies to analyze the impact of robust pretraining strategies and position embedding designs. In Table 3a, we progressively apply modifications including attentive pooling, and stronger data augmentation. Each contributes positively to K400 and SSV2, highlighting the importance of stable pretraining for video understanding. In Table 3b, we compare different designs for position embedding. We find that moving from absolute

| Data | K400 | SSV2 |
|---|---|---|
| Baseline | 64.1 | 15.0 |
| +Atten Pool | 65.2 | 15.5 |
| +Data Aug | **65.6** | **15.6** |

(a) **Pretraining Strategy**. The effects attentive pooling, and data augmentation.

| Method | K400 | SSV2 |
|---|---|---|
| Abs + ViT | 64.2 | 13.9 |
| Abs + 2DRoPE | 64.7 | 14.3 |
| M-RoPE | 65.1 | 15.2 |
| U-RoPE | **65.6** | **15.6** |

(b) **Position Embedding**. Comparison of absolute position, 2D, 3D, and unified RoPE strategies.

| Frames | K400 | SSV2 |
|---|---|---|
| Baseline | 62.7 | 13.2 |
| +8 | 64.4 | 14.2 |
| +4,8 | 65.2 | 15.3 |
| +1,2,4,8 | **65.6** | **15.6** |

(c) **Varing Frames with Multi-Event**. Pretraining at varying frames with multi-event.

| Method | K400 | SSV2 |
|---|---|---|
| only Obj. | 62.3 | 12.5 |
| only Evt. | 61.9 | 11.3 |
| Obj.+Evt. | 62.7 | 13.2 |
| Obj.+Multi-Evt. | **65.6** | **15.6** |

(d) **Clustering Strategy**. Perform clustering at the Event-level and Object-level.

| Num Classes | K400 | SSV2 |
|---|---|---|
| 500k | 65.2 | 15.5 |
| 1M | 65.6 | **15.6** |
| 2M | **65.8** | 15.5 |
| 5M | 65.2 | 15.4 |

(e) **Classes Number**. The number of classes during event-level and object-level clustering.

Table 3: **Ablation experiments** on K400 and SSV2 under few-shot settings. (a) Pretraining strategies. (b) Position embeddings. (c) Multi-event frame sampling. (d) Semantic clustering strategies. (e) Number of clustering classes. The entries marked in gray are the same, which specify the default settings.

position embedding to 2D-RoPE with 1d-absolutioe position and 3D-RoPE leads to substantial performance gains, with 3D-RoPE achieving better results. This suggests that temporal embedding better captures motion dynamics. Our U-RoPE decouples spatial and temporal dimensions during embedding, enabling more flexible handling of both image and video modalities, and achieves the best overall performance.

**Effect of Variable Spatiotemporal Streams.** We begin with a baseline where the model is pretrained using only 16-frame video clips. To enhance temporal modeling, we introduce variable-length frame inputs (*e.g.,* 1, 2, 4, 8, 16 frames) with dense labels, as shown in Table 3c. On K400, performance remains stable across frame combinations. In contrast, the more temporally complex SSV2 benefits notably from dense supervision with diverse frame counts, suggesting it helps capture short-range temporal dynamics. To qualitatively assess the short-range temporal robustness of our model, we visualize patch-level features using PCA, as shown in Fig. 6. The model produces consistent local semantic representations across a variety of image and video inputs, demonstrating strong spatial grounding and frame-invariant semantic encoding. Together, these findings validate the effectiveness of our variable-frame pretraining strategy and highlight the model's generalization capability.

**Qualitative Analysis of Frame-Agnostic Semantics.** We visualize the object-level feature distribution using T-SNE projection on K400, comparing both image and video samples. As shown in Fig. 7, our model produces well-formed and compact clusters, where image and video instances from the same semantic class are consistently grouped together. This indicates a strong alignment of visual representations across modalities. The tight intra-class clusters and clear inter-class boundaries demonstrate the model's ability to abstract high-level semantics that are shared between static and temporal visual data. Compared to the State of the Art model, such as SigLIP2 and DINOv2, our method achieves superior intra-class compactness and inter-class separation, highlighting its strength in learning modality-invariant and semantically consistent features.

**The Effect of Clustering.** To investigate the impact of different clustering strategies on video understanding, we perform ablation studies at both the event and object levels. As shown in Table 3d, using only object-level clustering achieves 62.3% on K400 and 12.5% on SSV2. Clustering only at the event level yields similar performance. However, combining object-level and event-level clustering leads to consistent improvements on both benchmarks (62.7% and 13.2%), indicating that the two levels provide complementary information. Furthermore, when we extend to multi-granularity event clustering (Obj.+Multi-Evt.), the performance improves further, achieving the best results on both datasets (65.6% and 15.6%). These results demonstrate that clustering at varying event granularities, in combination with object-level semantics, significantly enhances video understanding.

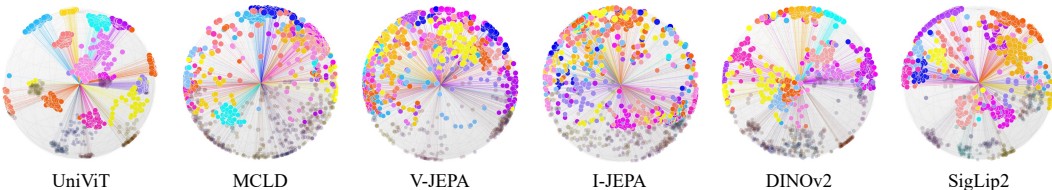

Figure 7: Visualization of object feature distributions using T-SNE projection onto a spherical space on K400 and K700 dataset. Class tokens serve as query vectors for QKV computation.

**Impact of Number of Class Choices.** As shown in Table 3e, we jointly vary the number of training classes. We observe that performance improves as both values increase, peaking at 1M classes. Interestingly, despite the significant increase in video data, the optimal number of classes aligns with that used in large-scale image pretraining, suggesting that current video datasets exhibit relatively low semantic diversity. At the same time, assigning multiple positives per sample proves beneficial for learning richer and more robust representations, especially in complex temporal tasks such as SSV2.

## 5 Related Work

**Advances in Visual Representation Learning.** The adoption of Vision Transformers [21, 39] has become a prevailing paradigm in the field of visual representation learning. Concurrently, equivariant self-supervised learning approaches [20, 45, 26, 29, 19] have emerged to predict structured data transformations consistent with group-theoretic formulations. Masked image modeling methods [30, 9, 22, 65] acquire visual representations by reconstructing masked regions of the input image in the pixel space. Moreover, JEPAs [6, 8] predict masked regions within a learned latent space rather than in the raw pixel domain. Contrastive Language-Image Pretraining (CLIP) [11, 55, 40, 24, 67, 50] aligns images and texts within a shared embedding space through instance-level contrastive supervision. However, existing approaches predominantly focus on either static image understanding or spatiotemporal modeling in isolation. In this work, UniViT structuring a shared semantic space for images and videos by modeling intra- and inter-instance, effective transfer to downstream tasks.

**Cluster Discrimination.** Instance discrimination methods [17, 31, 47], exemplified by CLIP [47, 69, 28], leverage instance-level contrastive supervision but neglect semantic similarities across instances, whereas cluster-based approaches [14, 5, 15] assign single pseudo-labels per sample, failing to adequately represent images containing multiple visual elements. To better capture semantic structures, cluster discrimination methods [14, 5, 72, 15] typically iteratively assign pseudo-labels through clustering and train classifiers based on these labels, grouping visually similar instances to encourage semantic coherence. However, conventional approaches assign only a single pseudo-label per image, limiting their capacity to represent multiple semantic concepts within one instance, an issue recently addressed by multi-label clustering methods such as Unicom [2] and MLCD [4]. In this work, we adopt multi-label clustering to unify the representation learning of images and videos, effectively enhancing the semantic coherence.

**Efficient Training.** Recent literature has explored various strategies for efficient CLIP training, such as large-batch optimization (up to 160K) [46, 50] and specialized optimizers like LAMB[56, 70]. RoPE originally designed for language models [54], has also been adapted to vision transformers via two-dimensional extensions [33, 1]. Additionally, significant efforts have focused on effective data curation and filtering at scale [25, 50, 24, 67], as well as image recaptioning using MLLMs [23, 38, 43, 66, 57, 3]. Motivated by these advances, we extend these methodologies to video data, constructing a unified data engine that facilitates robust representation learning across both images and videos.

## 6 Conclusion & Limitation

In this paper, we introduced UniViT, a cluster-driven unified self-supervised learning framework designed to jointly capture structured semantics across both spatial and temporal modalities through clustering and discrimination. Leveraging multi-granularity event-level clustering for videos and object-level clustering for images, UniViT first constructs structured semantic clusters across modalities during the clustering stage. Subsequently, in the discrimination stage, UniViT explicitly incorporates these clusters as pseudo-labels into a contrastive objective, effectively integrating structured semantic representations into a unified embedding space. To address limitations inherent to traditional

position encoding and fixed-input approaches, we further introduced U-RoPE and $VS^2$, enhancing semantic stability and flexibility across modalities. Extensive evaluations across multiple benchmarks demonstrate UniViT's superior scalability and its ability to achieve state-of-the-art performance on various downstream tasks, highlighting its effectiveness in unified visual representation learning.

**Limitation:** We clarify the limitations of our proposed UniViT: *(i):* UniViT relies on offline clustering with pretrained embeddings, potentially introducing biases from the initial feature extraction model and limiting its ability to adaptively update cluster assignments during training. *(ii):* Although our $VS^2$ strategy supports flexible temporal lengths and arbitrary input resolutions, long sequences or extremely high-resolution inputs may still incur substantial computational costs, potentially constraining practical scalability in resource-limited scenarios.

# 7   Acknowledgments

This work was supported by the Center of Excellence for Antimicrobial Therapeutics Discovery and Innovation (CEATDI), Grant No. 8002003.

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
