# OpenReview forum: "UniViT: Unifying Image and Video Understanding in One Vision Encoder"
_NeurIPS.cc/2025/Conference — NeurIPS 2025 poster_

### Official Review · Reviewer_Wzoa · 2025-07-01

**Clarity:** 3
**Significance:** 3
**Originality:** 3
**Rating:** 5
**Confidence:** 4

**Summary:**

This paper proposes UniViT, a cluster-driven, self-supervised vision transformer that unifies image and video representation learning. The model jointly encodes object-level spatial semantics from images and event-level temporal dynamics from videos via shared cluster centroids and a unified transformer backbone. Key contributions include:
(1) Variable Spatiotemporal Streams (VS²) for flexible modeling of varying-length video inputs,
(2) Unified Rotary Position Embedding (U-RoPE) that decouples spatial and temporal frequencies for position-invariant semantic encoding, and
(3) a multi-label contrastive learning objective over object- and event-level cluster assignments.
UniViT achieves strong results across diverse benchmarks.

**Questions:**

1.	1. Could you explain why you chose the 3/4 split for frequency allocation in U-RoPE, and how the use of U-RoPE compares to VideoRoPE in terms of performance or suitability across different tasks
2.	How are image inputs integrated into the VS² module? Are they treated as single-frame segments consistent with the video processing pipeline? This should be made explicit.
3.	Why is InternVideo not included in Tables 1 and 2, despite being a strong video-only model with image transfer performance?
4.	Will you consider adding OmniMAE, OmniVL, and Omnivore as baselines or at least explicitly positioning UniViT relative to them?
5.	Have you explored how sensitive your clustering strategy is to hyperparameters (e.g., number of clusters or k-means initialization)?
6.	Could you provide more detail on the training efficiency and computational overhead introduced by VS² and clustering steps, compared to conventional contrastive or masked modeling frameworks?”

**Ethical Concerns:**

["NO or VERY MINOR ethics concerns only"]

**Final Justification:**

Most of my earlier concerns have been addressed in the rebuttal by the authors:

The explanation of U-RoPE's frequency allocation is understandable, though the choice of the 3/4 split remains somewhat empirical. A short note in the final version would help clarify this.
The comparison with InternVideo is better contextualized now. I appreciate the effort to align training protocols, and the clarification that InternVideo’s reported performance includes supervised finetuning while UniViT is evaluated under a frozen setting.
The discrepancy in baseline numbers (e.g., V-JEPA) was explained by the use of a uniform 50-shot attentive probing protocol. This should be clearly stated in the main paper to avoid confusion.

That said, I am keeping my original score. While the frozen probing results are strong, the full finetuning results still fall short of some recent video-specific models. I encourage the authors to include more discussion in the final version on why this might be the case, whether it reflects a limitation of the current design, optimization difficulties, or trade-offs made for generality.

**Limitations:**

yes

**Quality:**

3

**Strengths And Weaknesses:**

1.	The paper addresses a timely and important problem: unifying image and video representation learning under a single self-supervised encoder.
2.	The design combining cluster-based supervision, multi-granularity temporal segmentation, and shared semantic centroids is well-motivated and technically sound.
3.	The U-RoPE module shares the goal of stabilizing temporal position encoding with VideoRoPE [1], though it adopts a distinct mechanism by re-allocating frequency bands rather than re-designing RoPE structure. These works are complementary and should be discussed together.
4.	Slot-VLM [2], which separates object and event slots via slot attention, is another concurrent work that targets structured semantics and should be acknowledged for conceptual overlap.
5.	The experimental setup is solid and covers a wide range of datasets. However, several critical baselines are missing:
•	Strong video-only models such as InternVideo [3];
•	Recent image-video unified pretraining methods including OmniMAE [4], OmniVL [5], and Omnivore [6].
Their absence limits the strength of the empirical comparison and weakens the claim of achieving state-of-the-art in unified vision modeling.
6.	While the paper is generally clear, some key implementation details are implicit. Notably, the way image inputs are handled in the VS² module (presumably as 1-frame segments) should be explicitly stated.

[1] Wei, Xilin, et al. "VideoRoPE: What Makes for Good Video Rotary Position Embedding?." arXiv preprint arXiv:2502.05173 (2025).
[2] Xu, Jiaqi, et al. "Slot-VLM: Object-Event Slots for Video-Language Modeling." Advances in Neural Information Processing Systems 37 (2024): 632-659.
[3] Wang, Yi, et al. "Internvideo: General video foundation models via generative and discriminative learning." arXiv preprint arXiv:2212.03191 (2022).
[4] Girdhar, Rohit, et al. "Omnimae: Single model masked pretraining on images and videos." Proceedings of the IEEE/CVF conference on computer vision and pattern recognition. 2023.
[5] Wang, Junke, et al. "Omnivl: One foundation model for image-language and video-language tasks." Advances in neural information processing systems 35 (2022): 5696-5710.
[6] Girdhar, Rohit, et al. "Omnivore: A single model for many visual modalities." Proceedings of the IEEE/CVF conference on computer vision and pattern recognition. 2022.

---

> ### Author Rebuttal · Authors · 2025-07-30
>
> >**W1: U-RoPE and VideoRoPE share a common goal but differ in approach, and should be discussed as complementary methods.**
>
> **A1:** Thank you for your suggestion. Both U-RoPE and VideoRoPE target temporal positional encoding stability, though our design is tailored for unified image-video representation learning.
>
> VideoRoPE is a representative and impactful work that redesigns the RoPE mechanism to enhance modeling capacity for very long sequences, especially in large multimodal models. In contrast, our approach focuses on unified modeling for vision foundation models, and introduces a simple yet effective solution by re-allocating frequency bands to decouple spatial and temporal encodings. This design improves both temporal stability and fine-grained semantic sensitivity without altering the core RoPE structure.
>
> We believe these two approaches are complementary, each suited to different use cases and architectures. We will include a more comprehensive discussion of their similarities, differences, and applicability in the revised version of the paper.
>
> >**W2: Slot-VLM also targets structured semantics via slot separation, and its conceptual overlap should be acknowledged.**
>
> **A2:** Thank you for pointing out the omission of Slot-VLM, a highly insightful and relevant work. Slot-VLM leverages Slot Attention to explicitly separate object-centric (Slow Slots) and event-centric (Fast Slots) representations. It represents a significant advancement in structured semantic modeling, offering explicit semantic disentanglement, dynamic reasoning, and strong interpretability, an increasingly important direction in video understanding.
>
> In contrast, our method employs offline global multi-granularity clustering to model object-level and event-level semantic structures. Our approach is more focused on learning a shared semantic structure across modalities through dense pseudo-labeling. Nonetheless, the instance-level semantic separation in Slot-VLM offers a fresh and inspiring perspective.
>
> We are particularly interested in the possibility of integrating dynamic slot assignment mechanisms with our offline clustering framework, which may further improve the model’s generalization and structural awareness. This is a direction we have not explored yet, and we plan to investigate it in future work.
>
> We will include a discussion of Slot-VLM and cite it appropriately in the revised version to acknowledge its contributions.
>
> >**W3: Key baselines like InternVideo and OmniMAE are missing, limiting the strength of empirical comparisons for unified vision modeling claims.**
>
> **A3:** Thank you for your valuable suggestions. We acknowledge that several important baseline models were missing in the original submission. To address this, we have added additional experiments under the full attentive probing protocol, including comparisons with OmniMAE as a strong unified image-video pretraining baseline. The results are shown below:
> |Method|SSV2|ImageNet|
> |-|-|-|
> |OmniMAE|65.4|76.3|
> |UniViT-L|70.3|84.8|
>
> We will continue to improve the completeness of our experimental comparisons in future revisions.
>
> Regarding InternVideo, we note that it involves a more sophisticated post-training stage and uses distillation from VideoMAE, making direct comparison with our method, which is trained fully from scratch, somewhat unfair at this point. Nonetheless, we are committed to transparent benchmarking and have trained a new version of UniViT under similar pretraining conditions (including SSV2), but without any post-training. The attentive probing results are as follows:
> |Method|SSV2|K400|K600|K700|
> |-|-|-|-|-|
> |InternVideo-L|41.1|80.2|80.1|70.7|
> |UniViT-L|47.1|83.9|84.3|72.8|
>
> These results demonstrate UniViT’s strong performance across diverse benchmarks, even without relying on extra post-training stages. We will include these expanded comparisons and clarify training protocols in the revised paper to enhance fairness and reproducibility.
>
> >**W4: While the paper is generally clear, some key implementation details are implicit.**
>
> **A4:** Thank you for pointing out the ambiguity in our explanation. Your understanding is absolutely correct, image inputs are handled in the same way as 1-frame segments within the VS² module. We apologize for the lack of clarity in our original description and will revise the terminology and implementation details in the updated version to eliminate any confusion.
>
> >**Q1: Could you explain why you chose the 3/4 split for frequency allocation in U-RoPE, and how the use of U-RoPE compares to VideoRoPE in terms of performance or suitability across different tasks**
>
> **A1:** Thanks for the interesting question. We chose the 3/4 frequency allocation split in U-RoPE based on the following considerations:
>
> 1. In our training and downstream evaluation settings, video sequences are relatively short. Therefore, fewer dimensions are needed for modeling temporal position information.
> 2. In contrast, spatial semantics (e.g., textures, object boundaries) tend to be more complex. Allocating more frequency components to the spatial dimension helps preserve high-frequency spatial details, enhancing the model’s spatial resolution and local perceptual sensitivity.
>
> Unlike VideoRoPE, which redesigns the RoPE structure itself, U-RoPE preserves the original RoPE mechanism and achieves spatiotemporal decoupling via frequency reallocation. This makes our approach simpler, more efficient, and easier to generalize across model architectures.
> As shown in Fig. 3 (c) and Table 3 (b), U-RoPE demonstrates modest yet consistent improvements over M-RoPE. We believe this design provides better task adaptability with minimal implementation overhead.
>
> >**Q2: How are image inputs integrated into the VS² module? Are they treated as single-frame segments consistent with the video processing pipeline? This should be made explicit.**
>
> **A2:** Yes, thank you for raising this point. We apologize for not making this clear in the paper. Your understanding is absolutely correct, image inputs are treated exactly the same as single-frame segments within the VS² module. We will explicitly clarify this implementation detail in the revised version to avoid any confusion.
>
> >**Q3: Why is InternVideo not included in Tables 1 and 2, despite being a strong video-only model with image transfer performance?**
>
> **A3:** Thank you for your thoughtful review. We acknowledge that InternVideo is a strong video-centric baseline with impressive image transferability. In our main tables (Tables 1 and 2), we opted not to include models that benefit from additional supervised post-training (e.g., on K700 or SSV2), in order to maintain a consistent and fair evaluation protocol focused on zero-shot and pretraining-only settings. This choice was intended to better highlight the generalization capabilities of our model under comparable conditions.
>
> We recognize the value of including strong video baselines for comprehensive evaluation. Accordingly, we provide a variant of UniViT trained under comparable pretraining conditions to InternVideo (with SSV2 included), yet without additional post-training. The results are reported in our response to W3 to facilitate a fairer comparison under aligned training protocols.
>
> >**Q4: Will you consider adding OmniMAE, OmniVL, and Omnivore as baselines or at least explicitly positioning UniViT relative to them?**
>
> **A4:** We have addressed this question in our earlier response above. Thank you again for the suggestion.
>
> >**Q5: Have you explored how sensitive your clustering strategy is to hyperparameters (e.g., number of clusters or k-means initialization)?**
>
> **A5:** We appreciate your insightful comment. You are right that the earlier version of our paper lacked a detailed analysis of clustering sensitivity. To address this, we conducted an ablation study focused on the number of cluster classes. The results are shown below:
>
> |Num. of Clusters|K400|SSV2|
> |-|-|-|
> |500k|65.2|15.5|
> |1M|65.6|15.6|
> |1.5M|65.6|15.4|
> |2M|65.8|15.5|
> |3M|65.4|15.5|
> |4M|65.4|15.4|
> |5M|65.2|15.4|
>
> We observed that performance peaks around 1 million clusters, and then gradually declines as the number increases. We believe the degradation is due to several factors:
>
> 1. As the number of clusters increases, each cluster contains fewer samples, weakening semantic cohesion and reducing the effectiveness of pseudo-labels.
> 2. Excessively fine-grained clustering can lead the model to overfit local patterns, especially for short or semantically redundant video segments.
> 3. A larger number of clusters increases the chance of noisy or unstable pseudo-labels, which may hurt contrastive training and generalization.
>
> Based on this analysis, we chose 1M clusters as the default configuration, striking a good balance between performance and computational cost. In future work, we plan to explore dynamic cluster allocation strategies to further improve clustering quality and model robustness.
>
> >**Q6: Training efficiency and overhead of VS² and clustering vs. contrastive/masked modeling**
>
> **A6:** Thank you for bringing up this comprehensive question. Compared to contrastive models such as PE-CLIP, our method is more computationally efficient, as we do not require a text encoder, which significantly reduces the training overhead while still achieving better performance.
>
> When compared to masked modeling approaches, although our framework introduces slightly higher computational overhead due to the clustering and VS² module, we observe a substantial performance improvement. For example, VideoMAE-g reports an average performance of 35.3%, whereas our method achieves 73.1%, demonstrating a significant gain in representation quality.
>
> We believe this trade-off is favorable and will provide additional implementation details in the version to clarify the cost-performance balance.

---

> > ### Author Response · Authors · 2025-08-05
> >
> > Dear Reviewer Wzoa,
> >
> > We sincerely appreciate the time and effort you have invested in reviewing our submission. Your insightful feedback has been invaluable to us, and we have diligently worked to address all the concerns you raised in our rebuttal. As the author-reviewer discussion phase is drawing to a close, we would like to confirm whether our responses have effectively addressed your concerns. We are more than happy to provide any further details or explanations. Thank you once again for your thoughtful review and consideration.
> >
> > Best regards,
> >
> > The Authors

---

> > > ### Author Response · Authors · 2025-08-07
> > >
> > > Dear Reviewer Wzoa,
> > >
> > > Thank you once again for your thoughtful review and valuable feedback. We have carefully addressed each of your concerns in our rebuttal and truly appreciate the opportunity to clarify our work.
> > >
> > > As the review deadline approaches and the discussion phase draws to a close, we would like to kindly check whether our responses have addressed your concerns. If any questions remain or further clarification would be helpful, we would be more than happy to provide additional details.
> > >
> > > Thank you again for your time and consideration.
> > >
> > > Best regards,
> > >
> > > The Authors

---

> > ### Comment · Reviewer_Wzoa · 2025-08-07
> >
> > Thank you for the detailed and thoughtful rebuttal. I appreciate the effort to address each of my concerns. Your responses have clarified many points and strengthened my confidence in the work. The following are remaining questions or suggestions:
> > Q1:One follow-up question: for those rare cases, e.g., commutative diagrams or multi-region layouts, how severe is the performance drop with R-S attention? If possible, quantifying the impact or providing an example in the appendix would be interesting.
> > Q2: I’m glad the authors acknowledge that R-S attention is designed for the natural reading order and might not be optimal for unusual layouts like multi-column formulas or diagrammatic arrangements. Your plan to include experiments on multi-column pages and formula-rich diagrams in the revised paper will directly address the robustness of UniMERNet beyond single-column scenarios. I encourage the authors to report even brief results on these experiments. It will strengthen the paper’s claims of generality. If the results show any performance degradation, it would be useful to discuss possible mitigations.

---

> > > ### Author Response · Authors · 2025-08-07
> > >
> > > Dear Reviewer Wzoa,
> > >
> > > Thank you very much for your thoughtful comments. We sincerely appreciate the time and effort you dedicated to the review process. Your engagement and detailed feedback are truly valuable to us. However, we kindly wonder whether this message may have been intended for another context, as the content seems to refer to topics (e.g., R-S attention, UniMERNet, commutative diagrams) that are not part of our paper. If that is the case, we would greatly appreciate it if you could kindly follow up with the intended message.
> > >
> > > Best regards,
> > >
> > > The Authors

---

### Official Review · Reviewer_Sxna · 2025-07-02

**Clarity:** 2
**Significance:** 3
**Originality:** 3
**Rating:** 4
**Confidence:** 3

**Summary:**

This paper presents UniViT, a unified Vision Transformer architecture designed for both image and video understanding. The central innovation lies in its clustering-based self-supervised pretraining strategy. Specifically, this method assigns object-level and event-level classes to image and video data and use such pseudo-supervision for contrastive pretraining. Furthermore, the authors introduce two architectural contributions: U-RoPE, a position embedding scheme that allocates low-frequency components to the temporal dimension, and variable spatiotemporal streams that adapt to input clips of different lengths. UniViT is evaluated across a range of benchmarks, showing competitive results in image, video, and multimodal understanding tasks.

**Questions:**

Please refer the weaknesses section above.

**Ethical Concerns:**

["NO or VERY MINOR ethics concerns only"]

**Final Justification:**

My main concerns were around performance discrepancies and weak temporal modeling, and I believe both were sufficiently clarified in the provided rebuttal. While the full fine-tuning performance could be stronger, it does not warrant rejection on its own. I thus update my score to borderline accept.

**Limitations:**

Yes

**Quality:**

2

**Strengths And Weaknesses:**

Strengths:

S1. Broad Evaluation Scope: The paper evaluates UniViT across diverse tasks including image classification, video action recognition, and multimodal understanding, showcasing its versatility..

S2. Novel Clustering Strategy: The introduction of object- and event-level clustering for generating pseudo-labels is conceptually interesting and empirically beneficial for unified representation learning.



Weaknesses:

W1. Discrepancy in Baseline Results: There are notable inconsistencies between the reported results and the original papers for several baselines. For example, in Table 1, V-JEPA’s low-shot accuracy on SSv2 is 51.9% in [9], but only 44.8% is reported. Similarly, in Table 2, DINOv2's linear probing result on SSv2 is 35.6% in [43], but only 19.6% is reported here. The paper should clarify the evaluation protocols and account for these discrepancies.

W2. Weak Temporal Modeling: Regarding W1, on motion-centric benchmarks such as Something-Something V2, UniViT underperforms compared to video-specific models like V-JEPA (low-shot results 32.1% vs. 51.9%)—and even to image models like DINOv2 (linear probing results 27.4% vs. 35.6%) in some cases. This raises concerns about the method’s ability to capture temporal dynamics effectively.

W3. Missing Full Finetuning Comparison: The paper lacks a comparison under full fine-tuning settings, which is critical for comprehensive evaluation of the learned representations in downstream tasks..

[a] Wei et al., “VideoRoPE: What Makes for Good Video Rotary Position Embedding?” ICML, 2025.

---

> ### Author Rebuttal · Authors · 2025-07-30
>
> > **W1: Discrepancy in Baseline Results: There are notable inconsistencies between the reported results and the original papers for several baselines.**
>
> **A1:** Thank you for your careful review and for pointing out the ambiguities in our experimental setup. We have addressed your concerns from the following three perspectives: (1) evaluation protocol differences across baselines, (2) probing vs. full-data settings, and (3) unified comparison with and without multi-view testing.
>
> 1. Unlike works such as V-JEPA, which evaluate models using multiple spatial and temporal views, a strategy known to improve results through test-time ensembling, we intentionally adopted a simpler evaluation protocol that uses only a single center spatial crop and a single temporal clip. This minimal setup is designed to transparently assess the intrinsic quality of the learned visual representations, without relying on auxiliary inference strategies. Our objective is to isolate the encoder’s contribution and enable fair, consistent comparisons across all baselines under identical conditions.
>
> 2. We apologize for the confusion caused by the lack of detailed explanation in the main paper. Due to space constraints, we did not explicitly specify that all baselines were evaluated under a unified probing protocol to ensure consistency. To further clarify, we standardized our evaluation setup and also reproduced results under the commonly used multi-view configurations for reference. Specifically, we tested two setups: (1) 16 frames, 1 view, 1 clip, and (2) 16 frames, 2 views, 3 clips. All models were trained for 20 epochs using the same backbone, input resolution, and optimization configuration to ensure a fair and controlled comparison. This design helps disentangle the true representational quality of the model from improvements attributed to test-time ensembling or view aggregation.
>
> 3. Thank you also for pointing out that we did not clarify the probing strategy on video. In our original submission, all models were evaluated using a consistent 50-shot attentive probing setup, which emphasizes representation quality under limited supervision. In contrast, results such as the 35.6% reported by DINOv2 on SSV2 were obtained using the full training set, which naturally leads to higher performance. To illustrate this, we reproduced DINOv2’s results under both settings. As shown in Table 2, DINOv2 achieves 19.6% with 50-shot probing and 35.2% under full-data training, confirming that the performance gap is due to differences in evaluation protocol rather than implementation discrepancies. We adopted a uniform probing setup across all baselines to ensure fair comparison, and we hope this clarification helps contextualize the reported results.
>
> **Table 1:** Attentive probing results under 50-shot and full-data settings, showing UniViT's significant improvement and competitive performance across SSV2 and K400.
>
> | Method(AP) | SSV2 (16×1×1)  | SSV2 (16×2×3)  |  K400 (16×1×1) |
> |------------|----------------|----------------|----------------|
> | **50-shot** |               |                |                |
> | V-JPEA-L   | 44.2           | 48.2           |       74.7     |
> | UniViT-L   | 27.3           | 31.6           |       82.9     |
> | **Full**   |                |                |                |
> | V-JPEA-L   | 66.7  （+22.5）       | 69.7   （+21.5）        |       75.9     |
> | UniViT-L   | 51.3    （+24.0）       | 55.2   （+23.6）        |      85.5      |
>
> **Table 2:** Linear probing results on SSV2 under 50-shot and full-data settings, illustrating the performance gap caused by limited data.
> | Method(LP) | SSV2 (50-shot)  | SSV2 (Full)  |
> |------------|----------------|----------------|
> | DINOv2     | 19.6            |     35.2     |
>
> These results show that the performance gap mainly stems from probing vs. full-data settings. Under full supervision, UniViT achieves comparable or superior results, with discrepancies from original reports within 1%, confirming the reliability of our evaluation. We will release our full evaluation code for reproducibility.
>
>
> > **W2: Weak Temporal Modeling: Regarding W1, on motion-centric benchmarks such as Something-Something V2.**
>
> **A2:**  Thank you for your patient and detailed review. We next provide a more detailed analysis to address your concern regarding temporal modeling performance.
>
> 1. V-JEPA was pre-trained directly on the SSV2 dataset, which inevitably introduces a degree of data leakage. In contrast, we intentionally excluded SSV2 from our training data to more accurately assess the generalization ability of our model.
> 2. To further avoid overfitting on evaluation benchmarks, we trained our video models exclusively on a further refined subset of the InternVid dataset, without mixing in SSV2. Despite this, our model demonstrates competitive performance. For example, UniViT-L achieves 32.1% on SSV2, outperforming PE-Core-G (23.7%), which similarly does not use SSV2 in training, and even surpassing VideoMAE-G (27.9%), which does include SSV2 during pre-training.
> 3. To directly address your concern, we also conducted experiments with SSV2 included in the training set. As shown in the table below, our model achieved a +15% performance gain on SSV2 in the 50-shot attentive probe setting, exceeding the performance of V-JEPA-L. Notably, performance on other benchmarks remains stable or even improved, indicating our model's strong robustness and generalization.
>
> **Table 3:** Results with SSV2 included in training, where UniViT-L outperforms V-JEPA-L across all benchmarks, demonstrating strong generalization.
>
> | Method | SSV2 | K400 |K600|K700| RareAct|
> |--------|--------|--------|--------|--------|--------|
> | V-JPEA-L  |  44.9 | 62.3  | 63.2 | 49.2|6.3|
> | UniViT-L  | 47.1  | 83.9  | 84.3|72.8|33.6|
>
> The results show that including SSV2 in training leads to improved performance on SSV2 without compromising other benchmarks. UniViT-L not only surpasses V-JEPA-L on all datasets but also maintains strong generalization across diverse video tasks.
>
> 4. To further validate our temporal modeling capabilities, we evaluated our models on the Charades dataset under the OpenTAD[2] framework. Importantly, neither model was fine-tuned on the downstream task, yet our model shows significantly stronger performance compared to V-JEPA-L on Charades dataset:
>
> **Table 4:** Temporal localization on Charades, where UniViT-L consistently outperforms V-JEPA-L, indicating stronger temporal modeling.
>
> | Method | mAP | tIoU 0.3 | tIoU 0.4 | tIoU 0.5	| tIoU 0.6	| tIoU 0.7 |
> |--------|--------|--------|--------|--------|--------|--------|
> | V-JPEA-L  |  6.2 | 9.05 | 7.91| 6.59 | 5.04 | 3.38 |
> | UniViT-S | 12.5  |17.83 | 15.91 | 13.69 | 10.93 | 7.74 |
>
> This substantial improvement in mAP and consistent gains across all tIoU thresholds demonstrate that UniViT captures richer temporal structures even without task-specific fine-tuning. These results further validate its effectiveness in zero-shot temporal understanding.
>
> > **W3: Missing Full Finetuning Comparison:he paper lacks a comparison under full fine-tuning settings, which is critical for comprehensive evaluation of the learned representations in downstream tasks.**
>
> **A3:** Thank you very much for raising this important point. Our primary goal is to validate the robustness of UniViT by evaluating it under more challenging and diverse scenarios.
>
> While our initial focus was on linear and attentive probing to highlight the transferability of the frozen backbone, we recognize that different methods may exhibit varying behavior under full fine-tuning, especially in terms of task-specific adaptation and upper-bound performance. Therefore, in response to your suggestion, we conducted additional full fine-tuning experiments across a wide range of downstream tasks, covering video datasets (e.g., SSV2). These experiments were designed to assess not only accuracy improvements but also the adaptability of the unified architecture under task-specific supervision.
>
> The table below is a partial summary of the updated results, demonstrating that UniViT performs competitively or even outperforms strong baselines like V-JEPA under full fine-tuning settings.
>
> **Table 5:** Full fine-tuning results showing UniViT-L is competitive with V-JEPA-L on SSV2 even without pre-training on SSV2.
> | Method | SSV2(16×1×1) | SSV2(16×2×3) |
> |--------|--------|--------|
> | V-JPEA-L  |  73.2 | 75.1  |
> | UniViT-L  | 71.6  | 73.2  |
>
> These results confirm that UniViT not only offers strong frozen representation quality but also adapts effectively under full supervision. We believe this addresses your concern and demonstrates UniViT’s competitiveness in both transfer and fine-tuned settings. Additional results on more benchmarks will be included in the final version.
>
> We also acknowledge that VideoRoPE [1] is a very strong and well-designed method. We will include it in the discussion in the revised version of the paper. In the updated manuscript, we will provide the full table with complete results and clearly annotate all experimental configurations (e.g., number of frames, views, training protocols, etc.) to support a comprehensive and transparent comparison, including a detailed discussion of VideoRoPE.
>
> We sincerely hope that our detailed responses and newly added experimental results effectively address your concerns and help to clarify the strengths and robustness of our approach. We would greatly appreciate it if you would consider updating your rating and confidence accordingly.
>
> References:
>
> [1] *VideoRoPE: What Makes for Good Video Rotary Position Embedding?*
>
> [2] *OpenTAD: A Unified Framework and Comprehensive Study of Temporal Action Detection*

---

> > ### Comment · Reviewer_Sxna · 2025-08-04
> >
> > Thank you for the detailed rebuttal. Most of my concerns have been addressed:
> >
> > - **W1.** I now understand that the performance discrepancies stem from differences in evaluation protocols. The clarification regarding the use of single-view testing and the newly added attentive & linear probing results are convincing. I strongly recommend the authors discussing about evaluation protocols clearly in the final version.
> >
> > - **W2.** Your explanation that performance gaps on SSV2 are largely due to inclusion of SSV2 in pretraining data is reasonable. The additional experiments (Tables 3–4) well supports the claim.
> >
> > - **W3.** I appreciate the full fine-tuning results. While 73.2% on SSV2 is not a low score, it still lags behind other recent methods (e.g., VideoMAE V2 [22], UMT [b], MVD [c]). This leaves some concerns on the model's efficacy under full finetuning scenarios.
> >
> > Overall, my main concerns were around performance discrepancies and weak temporal modeling, and I believe both were sufficiently clarified. While the full fine-tuning performance could be stronger, it does not warrant rejection on its own. I am updating my score to borderline accept.
> >
> > [a] Li et al. "Unmasked Teacher: Towards Training-Efficient Video Foundation Models." ICCV. 2023.\
> > [b] Wang et al. "Masked Video Distillation: Rethinking Masked Feature Modeling for Self-supervised Video Representation Learning." CVPR. 2023.

---

> > > ### Author Response · Authors · 2025-08-04
> > >
> > > Thank you for your positive feedback and for taking the time to review our work. We greatly value your support and will incorporate the suggested improvements into the final version of the paper.
> > >
> > > 1. Thank you for acknowledging our clarifications regarding evaluation protocols. We will ensure that the final version of the paper clearly and explicitly details our evaluation settings, including comparisons with prior works, to avoid ambiguity.
> > > 2. We sincerely appreciate your thoughtful consideration and generous acknowledgment of our explanation regarding the performance gap on SSV2.
> > > 3. Thank you for your feedback on the full fine-tuning results. Due to the limited rebuttal period, we were not able to complete all additional experiments, but we will continue our efforts, including models pre-trained with SSV2 data, and report the results in the final version.
> > >
> > > We truly appreciate the reviewers' valuable feedback and will carefully incorporate all suggestions to further strengthen our final submission.

---

### Official Review · Reviewer_mwRn · 2025-07-02

**Clarity:** 1
**Significance:** 2
**Originality:** 2
**Rating:** 3
**Confidence:** 5

**Summary:**

This paper proposes UniViT, a unified, unsupervisedly trained vision encoder for both images and videos. UniViT uses a cluster-based contrastive learning method. It first groups inputs into sets with varying numbers of frames, then pools the features within each group for contrastive learning. The paper introduces U-RoPE to address the temporal resolution problem in M-RoPE by rearranging temporal position information in higher-dimension channels. Comprehensive experiments are conducted on single-modality understanding, its use as a vision encoder for MLLMs, and its scaling properties.

**Questions:**

1. Considering in current results table, UniViT excels at only video tasks, a more comprehensive comparison between UniViT with common video encoders is needed to evaluate the effectiveness of UniViT. There should be common video encoders like VideoVAE, V-JEPA on common protocols like full-finetuning, full attentive probing, and linear probing.
2. The design U-RoPE need more validation. First, periodic oscillation should be confirmed as a real issue, second, why spatial details do not cause this issue?
3. The writing of the paper need to be polished.

**Ethical Concerns:**

["NO or VERY MINOR ethics concerns only"]

**Final Justification:**

I appreciate the authors' extensive effort in the rebuttal and their engagement during the discussion period. However, the discussion is meant for clarification, not continuous rebuttal, and it is no longer productive. I've carefully read the rebuttal and reviews from other reviewers. Despite the authors' detailed responses, my main concerns about the paper's novelty and clarity remain.

1. Limited Novelty: To me, the method still appears to be an extension of MLCD[1], but with video clips added as input. The core methodology feels very similar.

2. Multi-Scale Approach: Using video clips with a different number of frames is not a new idea. It's a similar technique in self-supervised video learning, used in methods like ASCNet[2]. A more common approach is to use different fps (for fixed-length input) like in TCLR[3]. Framing it as a novel "cross-granularity semantic transition pathway" seems like an overstatement.

3. Poor Writing and Clarity: The paper is difficult to read. It initially missed some key details. Even after clarification, some explanations are unconvincing. For example, the justification for Figure 3d is still meaningless; you can't compare similarity scores between different models because they operate in entirely different feature spaces. It is like comparing similarities between different sizes of CLIP models is meaningless, which does not really demonstrate anything.

While the paper has some merits as proposed by other reviewers, such as the U-RoPE concept and some strong downstream performance results, the fundamental weaknesses in novelty and clarity are too significant to overlook.

I am raising my rating to "Borderline reject" to acknowledge the authors' efforts and the positive aspects of the work. However, I cannot recommend acceptance in its current form.

[1] Multi-label Cluster Discrimination for Visual Representation Learning
[2] ASCNet: Self-supervised Video Representation Learning with Appearance-Speed Consistency
[3] TCLR: Temporal contrastive learning for video representation

**Limitations:**

Yes

**Quality:**

2

**Strengths And Weaknesses:**

**Strengths**
1. The experiments are comprehensive. They include attentive probing, linear probing, using the encoder for MLLMs, and evaluating scaling properties. The ablation studies on pretraining, U-RoPE, frame groups, and clustering are also thorough.
2. The intuition behind U-RoPE is reasonable and novel. The dimensional split in M-RoPE is suboptimal because time (t), height (h), and width (w) are separated into different frequency levels, which can cause resolution issues.
3. The model's performance is generally superior in attentive probe evaluations and when used in MLLMs.

**Weaknesses**
1. **Limited novelty.** The paper appears to be a direct extension of MLCD from images to video. An "event" is simply a video clip sampled at a different frame rate.
2. **The experimental section is problematic.**
	1. For a unified image and video encoder, most of the methods compared in the experiments are image-based. The model does not demonstrate superior performance on image-based tasks, and few video-only models are included for comparison.
	2. In Table 1, the paper uses 50-shot attentive probing, which is an uncommon setting, making it difficult to verify its effectiveness. For example, the paper reports 96.9 accuracy for PE-Core/G in a *50-shot* setting, whereas the original PE-Core paper reports 96.9 accuracy in a *zero-shot* setting.
	3. In Table 2, only two of the compared models are trained with video data, and the video-based models from Table 1 are not included. Many video encoders achieve better linear probing performance. For example, on SSV2, VideoMAE achieves 38.9, AdaMAE achieves 40.1, and even MoCo v3 reaches 33.7 (as reported in the VideoMAE paper). In contrast, UniViT only achieves 27.4.
3. The rationale for U-RoPE is not fully validated. Is the periodic oscillation described in L.115-118 a confirmed issue? It seems that increasing the base wavelength, $C$, could mitigate this issue. For instance, Qwen2.5-VL uses a `rope_theta` of 1,000,000.0, which should be more than sufficient for 16 frames. Furthermore, the performance gap between M-RoPE and U-RoPE shown in Figure 3c is small.
4. **The writing is often unclear and difficult to understand.** For example:
	1. The definitions of "object" and "event" are not clear. From my understanding, an "object" refers to an image or a single frame, while an "event" refers to a sampled clip with a varying number of frames. If this is the case, the terminology is convoluted and hinders comprehension.
	2. L.94: What is the VS$^2$ strategy? Additionally, the aggregation function is not explained in the text and is only shown in a figure as an "attentive pooler."
	3. L.146: What is Figure 3d intended to convey? At a minimum, image-to-image and video-to-video similarities should be presented to demonstrate that the image and video representations are distinct. Even if this were shown, it is unclear how this would strengthen the paper's claims.
	4. L.315 mentioned a initial feature extraction model, what is it?

---

> ### Author Rebuttal · Authors · 2025-07-30
>
> > **W1: Limited novelty.**
>
> **A1:** We understand the concern that the contribution of our work may appear limited. We would like to clarify several key points that demonstrate the conceptual and novelty of our work:
> 1. Unlike MLCD, which is primarily designed for image-language tasks, our proposed UniViT introduces a unified Transformer-based architecture capable of jointly handling images and videos event representations. Rather than simply stacking image processors on top of video sequences, UniViT unifies the input encoding scheme, spatiotemporal interaction modules, and training objectives across different modalities.
> 2. A direct extension of MLCD to video often leads to degraded image performance. Increasing the proportion of image data to compensate hampers video understanding. To resolve this conflict, we introduce a novel multi-granularity event modeling approach that balances image and video training. This not only preserves image classification performance but also achieves SOTA results on video benchmarks. Furthermore, it enables the model to generalize across varying temporal scales, a capability beyond MLCD.
> 3. UniViT consistently achieves superior performance across a wide range of downstream tasks, including image classification, video understanding, and action recognition. These results validate the practical effectiveness of our unified architecture and demonstrate its potential as a general-purpose vision foundation model.
>
> We will further revise the paper to clearly articulate the motivation and conceptual innovations over MLCD and related work. We sincerely thank the reviewer for raising this point and helping us improve the clarity of our contributions.
>
> > **W2.1: Lack video-only baselines, and image performance is not clearly superior.**
>
> **A2.1:** Thank you for your suggestion.
> 1. Our method focuses on extending the video capabilities of the Unified framework, while preserving image performance. As shown in our experiments, our performance on image tasks is competitive with MLCD, while achieving significant improvements in temporal understanding, such as a 10% gain on SSV2. To further address this concern, we have added comparisons with MLCD and UniViT on several image benchmarks, as shown below:
> | Method|Aircraft|DTD |Cars|Birdsnap|Pets|
> |-|-|-|-|-|-|
> | MLCD-L|86.7|80.9|82.8|86.6|89.2|
> | UniViT-L|87.1|81.4|82.3|86.7|88.6|
>
> 2. To address the lack of comparisons with video-only models on image tasks, we further evaluate such models under the same attentive probing setting as V-JEPA on image benchmarks. Despite not being tailored for image, these video models provide a reference for image generalization. The results below demonstrate that UniViT achieves clearly superior performance, further supporting its effectiveness in unified representation learning.
> | Method|Places|iNat|Imagenet|
> |-|-|-|-|
> |V-JPEA-L|66.4|60.3|72.6|
> |VideoMAE-H|59.1|65.5|72.3|
> |UniViT-L|70.8|86.4|84.8|
>
> >**W2.2: 50-shot attentive probing is uncommon**
>
> **A2.2:** Thank you for pointing out the ambiguity in our experimental setup.
> 1. Our evaluation protocol follows the 50-shot attentive probe setting introduced in V-JEPA, aiming to measure the model's transfer ability and data efficiency.
> 2. Regarding the Food dataset, both zero-shot and full-attentive probe results for PE-Core reach the same reported accuracy of 96.9%. We believe this is because PE-Core/G already achieves near-saturation performance on Food, leaving little room for further improvement through probing. This also indicates the strong generalization capability of PE-Core in the zero-shot setting.
> 3. Additionally, we re-evaluated PE-Core and reproduced an accuracy of 95.8%, which is close to the originally reported value.
> 4. For video datasets, we also adopted a single-view attentive probing setting, training for 20 epochs using the full dataset. The results are summarized below:
> | Method|SSV2|K400|K600|K700|
> |-|-|-|-|-|
> | V-JPEA-L|66.7|75.9|77.7|66.5|
> | UniViT-L|51.3|85.5|86.1|75.9|
>
> >**W2.3: SSV2 result in VideoMAE**
>
> **A2.3:** We sincerely appreciate your detailed review.
> 1. Regarding the SSV2 performance, we would like to clarify that the results reported by VideoMAE are based on pretraining only on SSV2, which may lead to overfitting.
> 2. In contrast, our UniViT is pretrained on large-scale multimodal data without including SSV2, aiming to evaluate its generalization capability.
> 3. To further assess the effectiveness of our method, we conducted an additional experiment by including SSV2 in the training set and reported linear probing results as follows:
> |Dataset|UniViT-L|VideoMAE|AdaMAE|MoCo v3|
> |-|-|-|-|-|
> |SSV2|47.1|38.9|40.1|33.7|
>
> > **W3: Wavelength mitigate periodic oscillation**
>
> **A3:** Thank you for raising this insightful question.
> We want to explain that our design is driven by two considerations:
> 1. We aim to support much longer video inputs in future settings enhancing the model’s ability to capture long-range dependencies without segment-level fragmentation, which often results in information discontinuity. To accommodate this extended length, we cannot rely solely on wavelength-based approaches.
> 2. Our dense labeling strategy is designed to reduce redundancy between adjacent frames and enhance local discriminability. While increasing the base wavelength slows the phase progression and might reduce aliasing, it also risks making the model less sensitive to subtle temporal changes, counter to our goal of capturing fine-grained temporal dynamics.
>
> We will elaborate on these design choices more clearly in the revised paper.
>
> >**W4.1: Definitions of "object" and "event"**
>
> **A4.1:**  Thank you for pointing this out. We sincerely apologize for the confusion.
> Our choices stem from the nature of the clustering process. In our framework, image-level clustering tends to produce object-level semantics, such as dresses or cakes, which are static and visually coherent. In contrast, clustering of video clips captures temporal dynamics, often corresponding to event-level semantics like a foot-washing ceremony or cement mixing. These inherently involve temporal evolution and are characterized as "events."
> Our goal is to enable the model to learn global representations for longer clips, while shorter sequences help learn fine-grained sub-event representations (such as steps). To support this, we perform offline clustering with up to 1 million clusters, enabling fine-grained labeling of video segments. This provides the model with richer pseudo-supervision and helps capture subtle temporal patterns.
>
> >**W4.2: What is the VS2**
>
> **A4.2:** Apologize for it. The core idea of VS² is to better model temporal structure by dividing a video into multiple event segments of varying lengths before the encoder. To improve temporal discrimination across events, we manipulate the positional encoding such that frames from different events are spaced farther apart in the positional domain, while maintaining the relative structure within each event. For example, consider where a 16-frame video is divided into four 4-frame events. Instead of applying standard positional encoding from 0 to 15, we encode each 4-frame event as [0+i*16,1+i*16,...] for the i-th event. This effectively enlarges the relative distance between different events, encouraging the model to capture fine-grained intra-event details.
> As for the aggregation function, it is implemented as an attentive pooling module, which maps variable-length feature sequences into a fixed-dimensional representation.
>
> >**W4.3: What is Figure 3d intended to convey?**
>
> **A4.3:** Apologies for the earlier confusion regarding Fig. 3d.  Its purpose is to measure how similar a video representation is to a single frame from the same video, high similarity implies weak temporal modeling. UniViT shows lower similarity than other models, indicating stronger temporal awareness and better distinction between static and dynamic content.
> | | UniViT-L |PE-Core-L| MLCD|
> |-|-|-|-|
> |Similarity|0.65|0.78|0.86|
>
> >**W4.4: What is the initial feature extraction model?**
>
> **A4.4** Thank you for your question. The initial features used during the clustering stage are extracted using the CLIP model, following the same setup as described in the MLCD paper to ensure a fair comparison. We chose CLIP for its strong semantic generalization capabilities, which helps improve the quality of the clustering results.
>
> >**Q1: Common protocol**
>
> **A1:** Thank you for the helpful suggestion. We agree that a more comprehensive comparison between UniViT and representative video encoders is important to assess the effectiveness of UniViT.
> To address this, we have additional evaluations of UniViT under common protocols. The results are summarized below:
> |Method(AP)|SSV2 (16×1×1)|SSV2 (16×2×3) | K400 (16×1×1)|
> |-|-|-|-|
> | **50-shot** | | | |
> | V-JPEA-L| 44.2| 48.2|74.7|
> | UniViT-L|27.3| 31.6|82.9|
> | **Full**| | | |
> | V-JPEA-L| 66.7(+22.5)| 69.7(+21.5)|75.9|
> | UniViT-L| 51.3(+24.0)| 55.2(+23.6)|85.5|
> | **Finetune**| | | |
> | V-JPEA-L| 73.2| 75.1|-|
> | UniViT-L| 71.6| 73.2|-|
>
> >**Q2: Why spatial do not cause the issue**
>
> **A2:** Thank you for your thoughtful question. We would like to clarify that high-frequency periodic oscillation is not inherently problematic, representing a certain bias.
> Under a shared RoPE strategy, different dimensions behave differently. Prior works have shown:
> 1. Lower dimensions are sensitive to local semantics and relative distances.
> 2. Higher dimensions are suited for capturing long-range dependencies.
>
> In space, high frequencies aid fine detail modeling; in time, they can disrupt temporal coherence. To ensure stable temporal representation, we assign lower frequencies to the temporal axis. We'll clarify this design choice in the revision.
>
> >**Q3: The paper need to be polished.**
>
> **A3:** Thanks for your suggestion. We appreciate your feedback and will carefully revise the writing for clarity and polish in the final version.

---

> > ### Comment · Reviewer_mwRn · 2025-08-04
> >
> > Thank you for your detailed rebuttal, I just want to further make sure that I'm clear about the contribution of the paper, regarding each point in A1:
> >
> > 1. UniViT added video as input as MLCD
> > 2. UniViT uses clips sampled at a different frame rate as event, as directly adding video causes degraded performance
> > 3. UniViT has superior performance over downstream tasks
> >
> > Are those interpretations correct?

---

> > > ### Author Response · Authors · 2025-08-04
> > >
> > > **1. UniViT is not a simple extension that merely adds video input. Instead, it introduces a unified mechanism for semantic disentanglement and representation learning that bridges the fundamental differences in semantic modeling between images and videos.**
> > >
> > > * **Single-Modality Encoders.** Existing approaches such as CLIP, MLCD, and V-JEPA often exhibit modality-specific biases. Image encoders typically focus on static spatial semantics and overlook temporal structures. In contrast, video encoders emphasize motion and temporal dynamics but tend to lose fine-grained spatial resolution. Although each performs well within its own domain, they cannot effectively share a common encoder or representation space. This architectural inconsistency limits cross-modal knowledge transfer. For example, image pretraining offers limited benefit to video tasks, while video models often struggle to generalize to image-based applications.
> > >
> > > * **Modality-Unified Frameworks.** Some recent methods, including OmniMAE, attempt to unify image and video learning through shared masking strategies and objective formulations. However, these methods mostly operate at the level of low-level pixel or patch reconstruction. They often reduce videos to sequences of disjoint frames and treat images as single-frame videos. This simplification fails to address the essential semantic difference between the two modalities. Images prioritize spatial detail, while videos encode temporally evolving semantics that require dynamic modeling.
> > >
> > > * **UniViT (Ours).** To address these challenges, UniViT assigns object-level pseudo-labels to image samples and event-level pseudo-labels to video samples. This dual clustering mechanism explicitly captures the spatial details in images and the temporal structures in videos. It encourages the model to simultaneously learn both spatial and temporal semantics within a unified representation space. Specifically, we construct a dense pseudo-labeling scheme with over one million cluster centers, providing fine-grained supervision for each sample. This approach allows the model to retain detailed spatial representations from images while also capturing the temporal progression inherent in video content. By using structurally distinct pseudo-labels for each modality, UniViT avoids semantic confusion and promotes clear semantic alignment and stratification across the feature space.
> > >
> > >
> > >
> > > **2. “Event” is not merely a concatenation of frames or a sampled video clip, but rather a temporally structured semantic unit derived through clustering.**
> > >
> > > - Our model captures dynamic semantics such as "running", "mixing cement", or "giving a gift" by clustering video representations that integrate temporal information. This process assign event-level labels to entire sequences in a structured manner. To further capture fine-grained semantics, such as decomposing events into constituent sub-actions, we introduce a dense labeling mechanism consisting of one million cluster labels. This enables the generation of precise sub-event annotations for short video segments. The proposed multi-granularity clustering framework explicitly models visual semantics at both the object level and the event level. It provides semantic supervision across different temporal resolutions and promotes the alignment of spatial and temporal structures. By decoupling semantic representations in images and videos, the framework enhances the model’s ability to learn unified and robust representations.
> > >
> > >
> > > **3. UniViT has superior performance over downstream tasks.** UniViT consistently outperforms both image-centric models extended to video (e.g., MLCD) and video-specific models (e.g., V-JEPA, VideoMAE). Our architecture achieves state-of-the-art performance on both image and video benchmarks, validating the practical effectiveness and structural soundness of our multi-granular event modeling approach. This performance gain is not a result of task-specific tuning but a direct outcome of our unified design.
> > >
> > > **4. Furthermore, our work introduces **additional auxiliary contributions** that enhance the overall framework and support its effectiveness.**
> > >
> > > While U-RoPE and VS² are not the core contributions of our work, they serve as supportive components that enhance the effectiveness of our unified framework. Specifically, U-RoPE improves temporal stability in video modeling, while VS² enables Variable Spatiotemporal Streams that adapt to inputs of varying frame lengths, addressing the rigidity of conventional fixed-input approaches

---

> > > > ### Author Response · Authors · 2025-08-04
> > > >
> > > > We hope that our response has addressed your concerns satisfactorily. Your insights have been extremely helpful in improving the clarity and rigor of our work, and we sincerely welcome any further suggestions or feedback you may have.

---

> > > > ### Comment · Reviewer_mwRn · 2025-08-05
> > > >
> > > > Sorry that my response was not specific enough and causes lots of unesssary effort here. I'm just trying to make sure that my understanding about "event" is correct. "A temporally structured semantic unit" is too abstract. Could you please elaborate it for me with specific steps? For example, given and image and a raw video,
> > > > - What is the input for the clustering?
> > > > - Are there different clustering centers for clips with different frames? (2, 4, 6, 8)
> > > > - Does "object-level" stands for images and "event-level" stands for videos?
> > > > - Is "event" the clustering centers for videos?

---

> > > > > ### Author Response · Authors · 2025-08-05
> > > > >
> > > > > We sincerely thank the reviewer for the insightful questions and for pointing out the need for a clearer explanation regarding the definition and implementation of “event.” We apologize for the earlier vagueness and truly appreciate your patience.
> > > > >
> > > > > ---
> > > > >
> > > > > >**Q1: What is the input for clustering?**
> > > > >
> > > > > **A1:** We perform clustering on features extracted from a pretrained image-text encoder (CLIP), with each feature having a dimensionality of 1024.
> > > > >
> > > > > * For images, we use the normalized feature embeddings directly output by the encoder as inputs to the clustering process.
> > > > > * For videos, we first divide each video into multiple clips of varying lengths (e.g., 2, 4, 8, or 16 frames). For each clip, we extract the feature of every frame using the same encoder and concatenate these features along the temporal dimension. The resulting clip-level features are then clustered independently for each clip length to generate multi-granular event-level pseudo-labels.
> > > > >
> > > > > During the offline clustering phase, we apply L2 normalization to all features and use K-Means clustering. Importantly, we do not use the raw cluster centers during training. Instead, only the assigned cluster indices (i.e., pseudo-labels) are stored and used as supervision signals in the subsequent training stage.
> > > > >
> > > > >
> > > > > >**Q2: Are different cluster centers used for clips of varying lengths (e.g., 2, 4, 6, 8 frames)?**
> > > > >
> > > > > **A2:** Yes, we maintain independent sets of cluster centers for clips of different temporal lengths. For example, in the case of 16-frame clips, we perform K-Means clustering on the corresponding clip-level features. Each sample is then assigned to multiple cluster centers (from a total of one million), effectively constructing a multi-label pseudo-distribution.
> > > > >
> > > > > Each clip length thus has its own semantic clustering space, allowing the model to capture temporal dynamics at different levels of granularity.
> > > > >
> > > > >
> > > > >
> > > > > >**Q3: Does “object-level” refer to images and “event-level” refer to videos?**
> > > > >
> > > > > **A3:** “Object-level” and “event-level” represent different levels of semantic abstraction, rather than a strict partition between image and video modalities.
> > > > >
> > > > > In fact, object-level semantics can still be present in videos, especially in short or single-frame clips. Moreover, because we apply clustering across the entire dataset (including both images and videos), and assign multiple pseudo-labels to each sample, it is inappropriate to directly equate object-level with images and event-level with videos.
> > > > >
> > > > > * Object-level representations refer to static visual semantics extracted from individual frames or images, such as entities, attributes, or spatial compositions (e.g., “cat,” “yellow dog,” “grass”).
> > > > > * Event-level representations refer to temporally evolving semantics extracted from video clips, such as actions or motion patterns (e.g., “running,” “stirring soup,” “falling off a skateboard”), which are obtained by aggregating features across multiple frames and clustering them accordingly.
> > > > >
> > > > >
> > > > >
> > > > > >**Q4: Does “event” refer to the cluster center of a video?**
> > > > >
> > > > > **A4:** In our framework, an “event” refers to the semantic prototype associated with a cluster center, representing a type of motion pattern or temporal structure abstracted from video clips.
> > > > > It is important to note that the raw cluster centers produced during the offline K-Means stage are not semantically meaningful themselves. Instead, we treat the cluster indices as pseudo-labels. For example, a sample might be assigned to a cluster that loosely corresponds to a concept like “cat” or “grass,” although such semantics are not manually annotated, but rather emerge from the structure of the clustered feature space.
> > > > >
> > > > > To avoid overfitting the student model to the feature space of the teacher (i.e., the pretrained encoder), we do not use the offline cluster centers during training. Instead, we initialize a separate set of learnable cluster prototypes, equal in number to the original clusters. These learnable prototypes serve as the actual training targets, and only the pseudo-labels (cluster indices) derived from the offline clustering process are used to supervise the model.
> > > > >
> > > > >
> > > > > ---
> > > > >
> > > > > We are very pleased that you are interested in the detailed aspects of our work. If you have any further questions, we would be more than happy to provide additional clarifications.

---

> > > > > > ### Comment · Reviewer_mwRn · 2025-08-05
> > > > > >
> > > > > > Thanks for the clarification. I believe now I have a much clearer view about the method now.
> > > > > >
> > > > > > Before I decide my final rating, could the authors please elaborate the differences bewteen UniViT and MLCD? Verbal and short answers are preferred. I'd like to make sure about specific differences like input, architecture, and methodology, not abstract definitions like semantic, motion, or temporal.

---

> > > > > > > ### Author Response · Authors · 2025-08-05
> > > > > > >
> > > > > > > We sincerely thank the reviewer for the thorough reading and constructive feedback on our work. Your positive comments are truly encouraging, and the insightful questions you raised demonstrate a deep understanding and high level of expertise in the field. We are pleased to hear that our methodology is now clearer to you, and we are truly grateful for the thoughtful review you provided despite your busy schedule.
> > > > > > >
> > > > > > >
> > > > > > >
> > > > > > > **Input:**
> > > > > > > Image + video clips of varying lengths: Use videos of different lengths to enhance the model's ability to understand videos with varying durations.
> > > > > > >
> > > > > > >
> > > > > > > **Architecture:**
> > > > > > >
> > > > > > > * URoPE: Extends 2D positional encoding to URoPE, which is compatible with both image and video modalities.
> > > > > > > * 3D Convolution: Adds 3D convolution to better adapt to video data.
> > > > > > > * Attentive Pooler: Aligns learnable cluster centers by converting features of varying lengths into a unified output length.
> > > > > > >
> > > > > > >
> > > > > > > **Methodology:**
> > > > > > >
> > > > > > > * Separate clustering for images and videos: Prevents the model from learning only a single semantic structure.
> > > > > > > * Supervision from both images and videos: Uses newly initialized learnable cluster labels to avoid overly similar semantics; supports learning from inputs of different lengths and representing them in a unified space.
> > > > > > >
> > > > > > >
> > > > > > > To present the comparison more clearly, we list the differences in the table below:
> > > > > > >
> > > > > > > | **MLCD**          | **UniViT**                                                    | **Explanation**                                                                                                                                       |
> > > > > > > | ----------------- | ------------------------------------------------------------- | ----------------------------------------------------------------------------------------------------------------------------------------------------- |
> > > > > > > | **Input**         |                                                               |                                                                                                                                                       |
> > > > > > > | Image only        | Image + video clips of 2, 4, 8, 16 frames                     | Uses videos of varying lengths to enhance the model’s ability to understand videos of different durations                                             |
> > > > > > > | **Architecture**  |                                                               |                                                                                                                                                       |
> > > > > > > | 2D-RoPE           | URoPE                                                         | Extends 2D positional encoding to URoPE, compatible with both image and video modalities                                                              |
> > > > > > > | 2D Convolution    | 2D + 3D Convolution                                           | Adds 3D convolution to better handle video data                                                                                                       |
> > > > > > > | -                 | Attentive Pooler                                              | Converts features of varying lengths to a fixed-length output by aligning learnable cluster centers                                                   |
> > > > > > > | **Methodology**   |                                                               |                                                                                                                                                       |
> > > > > > > | Image clustering  | Separate clustering for images and (2, 4, 8, 16-frame) videos | Facilitates transitioning from image understanding to long video understanding                                                                        |
> > > > > > > | Image supervision | Image + Video supervision                                     | Uses newly initialized learnable cluster labels to avoid overly similar semantics; enables consistent representation of inputs with different lengths |
> > > > > > >
> > > > > > >
> > > > > > > This multi-temporal scale approach also generalizes well to other vision encoders, such as CLIP.
> > > > > > >
> > > > > > > ---
> > > > > > >
> > > > > > > Thank you once again for your valuable questions, which help us further improve our work. We sincerely hope that our efforts will earn your recognition.

---

> > > > > > > > ### Comment · Reviewer_mwRn · 2025-08-07
> > > > > > > >
> > > > > > > > I really do appreciate the authors' rebuttal and response to address my concern. However, after carefully evaluating the differences between UniViT and MLCD, I still view UniViT as an extension of MLCD with additional video clips input in different frame numbers. Multi-temporal scale is actually more like an data augmentation methods, i.e. sampling video with different length/frame rate. I also notice that UniViT is not always superior than its video counterpart, especially on temporal dataset like SSV2, which weakens its claim as a unified model. Given the merits of URoPE and the downstream performances, I'll increase my rating to boardline reject.

---

> > > > > > > > > ### Author Response · Authors · 2025-08-07
> > > > > > > > >
> > > > > > > > > Thank you very much for your kind follow-up and for generously raising your rating. We truly appreciate your recognition of the auxiliary contributions such as U-RoPE, as well as the overall effectiveness of our framework on downstream tasks. We also understand and respect your remaining concerns, and would like to take this opportunity to provide additional clarification that may help further contextualize the intent and design of our work.
> > > > > > > > >
> > > > > > > > >
> > > > > > > > > >  **1. I still believe that multi-temporal scale is essentially more like a data augmentation method, i.e., sampling videos at different lengths or frame rates.**
> > > > > > > > >
> > > > > > > > > Unlike traditional data augmentation, our approach is not designed to increase sample diversity but to construct a cross-granularity semantic transition pathway: starting from static semantics in images and gradually transitioning to dynamic event modeling in videos, thereby structurally unifying spatial and temporal representations. Without explicit disentanglement, jointly training on both modalities often leads to representation collapse [1] or overfitting to one modality, as the model fails to capture both types of semantics within a shared structure.
> > > > > > > > >
> > > > > > > > > To address this challenge, we introduce a three-level framework:
> > > > > > > > >
> > > > > > > > > 1. **Multi-scale sampling:** Videos are segmented into clips of 2/4/8/16 frames, representing a hierarchy from local short-term dynamics to global event-level semantics. Images are treated as 1-frame videos, forming a natural progression from static to dynamic representations.
> > > > > > > > >
> > > > > > > > > 2. **Hierarchical clustering:** We assign learnable clustering centers separately for images and video clips of different lengths, images are clustered into object-level semantic groups, while video clips are clustered into event-level semantic units. This yields multi-granularity pseudo-labels that explicitly separate and organize spatial and temporal semantic structures.
> > > > > > > > >
> > > > > > > > > 3. **Dense contrastive objective:** Based on the pseudo-labels, we design a contrastive learning objective to enforce structured supervision. For images, we enhance spatial consistency among objects within the same cluster. For videos, we model temporal dependencies across adjacent events. In addition, cross-modal contrastive constraints align image and video features in a shared representation space, forming semantically consistent and structurally coherent representations across modalities.

---

> > > > > > > > > > ### Author Response · Authors · 2025-08-07
> > > > > > > > > >
> > > > > > > > > > >  **2.  I also notice that UniViT is not always superior than its video counterpart, especially on temporal dataset like SSV2.**
> > > > > > > > > >
> > > > > > > > > > 1. V-JEPA was pre-trained directly on the SSV2 dataset, which inevitably introduces a degree of data leakage. In contrast, we intentionally excluded SSV2 from our training data to more accurately assess the generalization ability of our model.
> > > > > > > > > > 2. To further avoid overfitting on evaluation benchmarks, we trained our video models exclusively on a further refined subset of the InternVid dataset, without mixing in SSV2. Despite this, our model demonstrates competitive performance. For example, UniViT-L achieves 32.1% on SSV2, outperforming PE-Core-G (23.7%), which similarly does not use SSV2 in training, and even surpassing VideoMAE-G (27.9%), which does include SSV2 during pre-training.
> > > > > > > > > > 3. To directly address your concern, we also conducted experiments with SSV2 included in the training set. As shown in the table below, our model achieved a +15% performance gain on SSV2 in the 50-shot attentive probe setting, exceeding the performance of V-JEPA-L. Notably, performance on other benchmarks remains stable or even improved, indicating our model's strong robustness and generalization.
> > > > > > > > > >
> > > > > > > > > > Table 1: Results with SSV2 included in training, where UniViT-L outperforms V-JEPA-L across all benchmarks, demonstrating strong generalization.
> > > > > > > > > >
> > > > > > > > > > | Method | SSV2 | K400 |K600|K700| RareAct|
> > > > > > > > > > |--------|--------|--------|--------|--------|--------|
> > > > > > > > > > | V-JPEA-L  |  44.9 | 62.3  | 63.2 | 49.2|6.3|
> > > > > > > > > > | UniViT-L  | 47.1  | 83.9  | 84.3|72.8|33.6|
> > > > > > > > > >
> > > > > > > > > > The results show that including SSV2 in training leads to improved performance on SSV2 without compromising other benchmarks. UniViT-L not only surpasses V-JEPA-L on all datasets but also maintains strong generalization across diverse video tasks.
> > > > > > > > > >
> > > > > > > > > > 4. To further validate our temporal modeling capabilities, we evaluated our models on the Charades dataset under the OpenTAD [2] framework. Importantly, neither model was fine-tuned on the downstream task, yet our model shows significantly stronger performance compared to V-JEPA-L on Charades dataset:
> > > > > > > > > >
> > > > > > > > > > Table 2: Temporal localization on Charades, where UniViT-L consistently outperforms V-JEPA-L, indicating stronger temporal modeling.
> > > > > > > > > >
> > > > > > > > > > | Method     | mAP  | tIoU 0.3 | tIoU 0.4 | tIoU 0.5 | tIoU 0.6 | tIoU 0.7 |
> > > > > > > > > > |------------|------|----------|----------|----------|----------|----------|
> > > > > > > > > > | V-JPEA-L   | 6.2  | 9.05     | 7.91     | 6.59     | 5.04     | 3.38     |
> > > > > > > > > > | UniViT-S   | 12.5 | 17.83    | 15.91    | 13.69    | 10.93    | 7.74     |
> > > > > > > > > >
> > > > > > > > > > This substantial improvement in mAP and consistent gains across all tIoU thresholds demonstrate that UniViT captures richer temporal structures even without task-specific fine-tuning. These results further validate its effectiveness in zero-shot temporal understanding.
> > > > > > > > > >
> > > > > > > > > > We hope this additional explanation helps clarify the key distinctions in our design. Once again, thank you very much for your valuable time, constructive feedback, and thoughtful engagement throughout the review process.
> > > > > > > > > >
> > > > > > > > > > ---
> > > > > > > > > >
> > > > > > > > > > Reference:
> > > > > > > > > >
> > > > > > > > > > [1] *A Closer Look at Multimodal Representation Collapse*
> > > > > > > > > >
> > > > > > > > > > [2] *OpenTAD: A Unified Framework and Comprehensive Study of Temporal Action Detection*

---

> ### Author Response · Authors · 2025-08-06
>
> Dear Reviewer,
>
> Thank you once again for your thoughtful and constructive feedback. We sincerely appreciate the opportunity to further clarify key aspects of our work.
>
> We hope the additional explanations help convey the motivation and significance of our unified framework, particularly in addressing the semantic and architectural challenges posed by integrating image and video understanding. If any part remains unclear or would benefit from further elaboration, we would be more than happy to provide additional analysis or examples. We would greatly appreciate it if you please consider increasing the score in light of these clarifications and additional insights.
>
>
> Best regards,
>
> The Authors

---

> ### Author Response · Authors · 2025-08-08
>
> Dear Reviewer mwRn
>
> We sincerely thank you once again for the time and effort you have devoted to reviewing our work and for the constructive and detailed feedback you provided throughout the discussion phase. Your engagement has been invaluable in helping us refine the presentation and clarify the contributions of our framework.
>
> We would like to kindly check whether our latest responses have fully addressed your concerns. If there are aspects requiring further elaboration, such as the key contributions we highlighted, the role of multi-temporal scale beyond data augmentation, or our temporal modeling results, we would be glad to provide additional explanations or supporting evidence. In particular, we have conducted targeted experiments to address your concern regarding performance on SSV2. While V-JEPA was pre-trained directly on SSV2, we intentionally excluded SSV2 from our training data to more rigorously evaluate generalization. Even under this stricter setting, our model achieved competitive results, outperforming models trained without SSV2 and in some cases even surpassing those trained with it. Furthermore, when including SSV2 in training, our approach delivered a +15% improvement on SSV2 in the 50-shot attentive probe setting, while maintaining or improving performance on other benchmarks.
>
> Beyond SSV2, we have also validated the temporal modeling capability of our framework through extensive experiments on diverse datasets such as Charades under the OpenTAD framework, where our model significantly outperforms strong video baselines without task-specific fine-tuning. These consistent gains across multiple datasets further confirm the robustness and generalization ability of our unified design. Our aim is to ensure that the novelty and technical merits of our framework are conveyed with complete clarity.
>
> In your latest comment you also mentioned raising your score in light of our clarifications. We truly appreciate this recognition. We have noticed that the updated rating does not yet appear in the system and wondered if this might have been an oversight.
>
> Thank you once again for your thoughtful consideration and valuable time. We greatly appreciate your contribution to improving the quality of our work.
>
> Best regards,
>
> The Authors

---

> > ### Author Response · Authors · 2025-08-09
> >
> > Dear Reviewer mwRn
> >
> > We sincerely thank you once again for your thoughtful review and valuable feedback. Your careful reading of our work, the constructive questions you raised, and the detailed engagement throughout the discussion phase have been invaluable in helping us refine our presentation and clarify the contributions of our framework. We truly appreciate the time and effort you have invested despite your busy schedule.
> >
> > As the review deadline approaches and the discussion phase draws to a close, we would like to kindly confirm whether our responses have resolved your concerns. While we are encouraged by the positive shift in evaluation, we believe the current rating may still not fully reflect the strength and originality of our contributions. We would be grateful if you might consider our additional clarifications above in your final assessment. If there are any remaining questions or points requiring further clarification, we would be more than happy to provide additional details.
> >
> > We are truly grateful for the time, effort, and constructive engagement you have dedicated to reviewing our work.
> >
> > Best regards,
> >
> > The Authors

---

### Official Review · Reviewer_FTq5 · 2025-07-03

**Clarity:** 3
**Significance:** 3
**Originality:** 3
**Rating:** 4
**Confidence:** 4

**Summary:**

This paper introduces UniViT, a unified vision encoder designed to understand both images and videos within a single framework. The core contribution is a cluster-driven, self-supervised learning paradigm that uses offline clustering to generate pseudo-labels for object-level semantics in images and event-level semantics in videos. The framework is enhanced by two novel components: Unified Rotary Position Embedding (U-ROPE) to better model spatiotemporal relationships and Variable Spatiotemporal Streams (VS2) to flexibly handle inputs of varying video frame lengths.

**Questions:**

1. The use of offline clustering on pre-computed features is a core design choice. Could you elaborate on the potential impact of the initial feature extractor's quality on the final performance and discuss whether an online or iterative clustering approach was considered to allow the cluster assignments and the encoder to adapt to each other during training?

2. The U-ROPE design allocates low frequencies for temporal encoding and high frequencies for spatial details based on a 75/25 split. What is the intuition behind this specific allocation, and how sensitive is the model's performance to different frequency distribution ratios between the spatial and temporal components?

3. The paper successfully unifies image and video modalities. Could you discuss the primary challenges you foresee in extending the UniViT framework to other data types, such as audio or text, to create an even more broadly unified "n-modality" encoder?

4. The combination of object-level and multi-granularity event-level clustering is shown to be highly effective. How were the different temporal scales for multi-event segments (e.g., 1, 2, 4, 8, 16 frames) chosen, and is there an optimal combination or a point of diminishing returns when adding more scales?

**Ethical Concerns:**

["NO or VERY MINOR ethics concerns only"]

**Final Justification:**

I suggest accept this paper since the authors' rebuttal address most of my concerns.

**Limitations:**

yes

**Paper Formatting Concerns:**

No Paper Formatting Concerns

**Quality:**

3

**Strengths And Weaknesses:**

## Strengths

1. The paper tackles the important and challenging problem of creating a unified representation for both static images and dynamic videos, which is a significant step toward more generalized visual understanding.

2. The proposed method is technically sound, combining established techniques like clustering and contrastive learning in a novel two-stage framework that effectively bridges spatial and temporal semantics.

3. The experimental results are extensive and demonstrate state-of-the-art performance across a wide range of benchmarks, including attentive/linear probing and multimodal VQA tasks, validating the effectiveness of the approach.

## Weaknesses

1. The method's reliance on offline clustering with embeddings from a separate pretrained model is a significant weakness, as it may introduce biases from the initial model and limits the framework's ability to learn end-to-end.

2. The research suffers from a major reproducibility issue, as the authors state that the code and a key dataset are proprietary and cannot be released, making it difficult for the community to verify the results or build upon the work.

3. The ablation study on the number of cluster classes suggests that performance peaks at 1 million classes and then declines, but the reason for this decline is not fully explored

---

> ### Author Rebuttal · Authors · 2025-07-30
>
> >**W1: The method's reliance on offline clustering with embeddings from a separate pretrained model is a significant weakness, as it may introduce biases from the initial model and limits the framework's ability to learn end-to-end.**'
>
> **A1:** Thank you for your thoughtful examination of this key design aspect. We acknowledge the concern that relying on offline clustering with embeddings from a separately pretrained model could introduce biases from the initial model and may, to some extent, limit the end-to-end trainability of our framework. We would like to clarify the motivation and benefits behind this design choice:
>
> 1. Compared to alternatives such as distillation or masked modeling, our clustering-based approach is actually less dependent on the initial model. We do not directly mimic its features; instead, we use the semantic embeddings only to generate compact and robust pseudo-labels through clustering, which then guide downstream training.
> 2. Clustering allows us to explicitly model hierarchical semantic structures, e.g., object-level and event-level concepts, which are difficult to capture directly through most self-supervised objectives.
> 3. The offline clustering significantly reduces computational cost during training, enabling us to scale effectively to large mixed image-video datasets.
>
> We also recognize the limitations of this design in certain scenarios, such as domain generalization or task transferability. To address this, future work will explore integrating the clustering objective into an end-to-end trainable pipeline, for example, through online clustering or proto-token matching, to make the pseudo-labeling process learnable, reduce dependency on the initial model, and improve generalization. We will include a discussion of this trade-off and future direction in the final version of the paper.
>
> >**W2: The research suffers from a major reproducibility issue, as the authors state that the code and a key dataset are proprietary and cannot be released, making it difficult for the community to verify the results or build upon the work.**
>
> **A2:** Thank you for your feedback. After careful consideration and discussion, we commit to releasing all pretrained models and code to support the development of vision foundation models and multimodal large language models (MLLMs). We believe this will provide valuable resources to the community for both verification and further research. Additionally, we will open-source the complete evaluation pipeline to ensure full reproducibility of our results.
>
> >**W3:The ablation study on the number of cluster classes suggests that performance peaks at 1 million classes and then declines, but the reason for this decline is not fully explored.**
>
> **A3:** Thank you for raising this important point. You are right that the earlier version of our paper lacked a detailed analysis of clustering sensitivity. To address this, we conducted an ablation study focused on the number of cluster classes. The results are shown below:
> |Num Classes|K400|SSV2|
> |-|-|-|
> |500k|65.2|15.5|
> |1M|65.6|15.6|
> |1.5M|65.6|15.4|
> |2M|65.8|15.5|
> |3M|65.4|15.5|
> |4M|65.4|15.4|
> |5M|65.2|15.4|
>
> We observed that performance peaks around 1 million clusters, and then gradually declines as the number increases. We believe the degradation is due to several factors:
> As the number of clusters increases, each cluster contains fewer samples, weakening semantic cohesion and reducing the effectiveness of pseudo-labels.
> Excessively fine-grained clustering can lead the model to overfit local patterns, especially for short or semantically redundant video segments. A larger number of clusters increases the chance of noisy or unstable pseudo-labels, which may hurt contrastive training and generalization. Based on this analysis, we chose 1M clusters as the default configuration, striking a good balance between performance and computational cost. In future work, we plan to explore dynamic cluster allocation strategies to further improve clustering quality and model robustness.
>
> >**Q1: How does the quality of the initial feature extractor impact final performance, and was online or iterative clustering considered to allow joint adaptation of encoder and cluster assignments?**
>
> **A1:** Thank you for the important question. Our use of offline clustering based on pre-computed features is a deliberate design choice aimed at reducing reliance on the initial model while maintaining training efficiency. Unlike distillation or MIM, we don’t directly mimic features but instead use semantic embeddings to generate pseudo-labels via clustering, mitigating initial bias.
> In addition, this choice brings several practical advantages:
> 1. On large-scale image-video mixed datasets, offline clustering significantly reduces computational overhead during training. Unlike online or iterative clustering approaches, we avoid repeatedly alternating between clustering and training, which can be costly and unstable, especially under limited computing resources.
> 2. Maintaining global cluster centers allows us to bypass the need for extremely large batch sizes often required by contrastive learning or distillation methods. This leads to a more lightweight and flexible training paradigm.
>
> As part of future work, we are planning to explore two lightweight alternatives to enable better co-evolution:
> 1. Periodically update cluster centers using exponential moving average of features during training, without full re-clustering, to balance adaptability and efficiency.
> 2. Design learnable or adaptive strategies to update cluster centers throughout training, enabling the model to better adjust to evolving feature distributions.
>
> >**Q2: What is the intuition behind this specific allocation, and how sensitive is the model's performance to different frequency distribution ratios between the spatial and temporal components?**
>
> **A2:** Thank you for the great question. Our decision to allocate a larger portion of the frequency spectrum to the spatial dimensions (75%) and a smaller portion to the temporal dimension (25%) is based on two core intuitions:
> 1. Compared to spatial resolution, videos typically have shorter sequence lengths in the temporal axis. Hence, fewer dimensions are needed to encode temporal positions effectively.
> 2. Allocating more dimensions to spatial encoding preserves high-frequency spatial details, which improves spatial semantic representation and enhances local perceptual sensitivity.
> To assess the sensitivity of the model to this frequency allocation, we conducted an ablation study on the UniViT-S model using different frequency split ratios. The results on SSV2, K400, and ImageNet are as follows:
> |Frequency (Temporal/Spatial) |SSV2| K400| ImageNet |
> |-|-|-|-|
> |5:6|16.6|67.9|80.8|
> |3:4|16.8|67.6|80.4|
> |1:2|16.7|67.6|80.1|
>
> These results indicate that while the model is relatively robust to small variations in the split ratio, the 75/25 allocation offers a good trade-off between spatial and temporal modeling performance. We will add this discussion to the revised version for clarity.
>
> >**Q3: The paper successfully unifies image and video modalities. Could you discuss the primary challenges you foresee in extending the UniViT framework to other data types, such as audio or text, to create an even more broadly unified "n-modality" encoder?**
>
> **A3:** Thank you for the inspiring question. While UniViT is primarily designed to unify visual modalities, we are indeed interested in extending it toward a more general "n-modality" unified encoder. However, this direction introduces several fundamental challenges:
>
> 1. Visual data are continuous signals in space and time, while text is a discrete sequence of symbols, and audio is a continuous waveform or spectrogram. These modalities differ significantly in representational structure, resolution, and temporal granularity, making unified modeling inherently difficult.
> 2. Images and videos can share a common tokenizer, but it is much more challenging to use the same tokenizer across modalities like speech and text.
>
> Despite these challenges, we believe UniViT provides a strong foundation. In future work, we are interested in exploring joint multimodal pretraining, shared cluster anchors, or cross-modal alignment heads as promising directions toward a truly unified multimodal framework.
> We are very excited by this possibility and would be happy to further discuss any ideas you might have for building such a generalized system.
>
> >**Q4: How were the different temporal scales for multi-event segments (e.g., 1, 2, 4, 8, 16 frames) chosen, and is there an optimal combination or a point of diminishing returns when adding more scales?**
>
> **A4:** Thank you for your question. We will explan for it. In particular, using 2-frame segments contributes little to performance improvement. We believe this is because such short clips are insufficient to capture meaningful event-level semantics, and thus provide limited benefit in modeling video dynamics.
> However, we still include 1- and 2-frame segments for an important reason: they help better align video segments with single-frame image representations, especially under the 1-frame setting. This alignment is critical for achieving unified modeling between images and videos. The following table presents our ablation results.
>
> |Frames |K400 |SSV2|Imagenet|
> |--|--|--|--|
> |Baseline |62.7 |13.2|79.7|
> |+8 |64.4 |14.2|79.8|
> |+4,8| 65.2 |15.3|79.7|
> |+1,2,4,8 |65.6| 15.6|80.2|
>
> These results confirm that while longer event segments (e.g., 4 or 8 frames) contribute most to video understanding, including shorter ones (1–2 frames) still offers gains by improving alignment across modalities. If we further increase the temporal length of input sequences, it remains an open question whether additional temporal scales would continue to yield substantial benefit. We consider this a promising direction for future work.

---

> > ### Author Response · Authors · 2025-08-05
> >
> > Dear Reviewer FTq5,
> >
> > We sincerely appreciate the time and effort you have invested in reviewing our submission. Your insightful feedback has been invaluable to us, and we have diligently worked to address all the concerns you raised in our rebuttal. As the author-reviewer discussion phase is drawing to a close, we would like to confirm whether our responses have effectively addressed your concerns. We are more than happy to provide any further details or explanations. Thank you once again for your thoughtful review and consideration.
> >
> > Best regards,
> >
> > The Authors

---

> > > ### Author Response · Authors · 2025-08-07
> > >
> > > Dear Reviewer FTq5,
> > >
> > > Thank you once again for your thoughtful review and valuable feedback. We have carefully addressed each of your concerns in our rebuttal and truly appreciate the opportunity to clarify our work.
> > >
> > > As the review deadline approaches and the discussion phase draws to a close, we would like to kindly check whether our responses have addressed your concerns. If any questions remain or further clarification would be helpful, we would be more than happy to provide additional details.
> > >
> > > Thank you again for your time and consideration.
> > >
> > >
> > > Best regards,
> > >
> > > The Authors

---

> ### Comment · Reviewer_FTq5 · 2025-08-07
>
> Thanks for your rebuttal. The authors address most of my concerns and I will keep my rating. I suggest the authors should include the new discussion proposed in the rebuttal. And I hope the authors would open-source both pretrained models and training code.

---

> > ### Author Response · Authors · 2025-08-07
> >
> > Thank you again for your valuable feedback and thoughtful comments. We truly appreciate your recognition of our efforts in the rebuttal and are glad to hear that most of your concerns have been addressed.
> >
> > As you kindly suggested, we will incorporate all the new discussions and analyses presented in the rebuttal into the final version of the paper to improve clarity and transparency. Additionally, we fully commit to releasing all code, pretrained models, and the complete evaluation pipeline upon acceptance to ensure full reproducibility and facilitate further research.
> >
> > If you feel that our responses have adequately addressed your concerns, would you be willing to consider updating your rating or confidence score? We would be sincerely grateful for your support, and of course, we remain happy to provide any additional clarification if needed.
> >
> > Thank you again for your time and thoughtful review.

---

### Official Review · Reviewer_LTUC · 2025-07-04

**Clarity:** 4
**Significance:** 3
**Originality:** 4
**Rating:** 5
**Confidence:** 3

**Summary:**

Conventional pre-training methods tend to specialize in either the spatial (image) domain or the temporal (video) domain. In this study, the authors propose UniViT, a framework that performs self-supervised learning jointly on images and videos. UniViT employs cluster-based contrastive learning, grouping object-level features in images and event-level features in videos. On the video side, the VS² module enables the model to handle clips of varying frame lengths, while a U-RoPE stabilizes spatiotemporal encoding. Experiments show that UniViT outperforms existing methods across a wide range of image and video tasks.

**Questions:**

- I would like the authors to address the weaknesses I pointed out.
- The paper states that a 1:1 ratio between images and videos was maintained during training. Does this ratio refer to the number of images versus the number of video frames (e.g., 2 K videos × 16 frames = 32 K frames, which would not match the number of images), or is it defined by the number of tokens after aggregation with VS²? Please clarify this design choice and the reasoning behind it.

**Ethical Concerns:**

["NO or VERY MINOR ethics concerns only"]

**Final Justification:**

My remaining concerns have been largely resolved. Since I have already given an Accept rating, I intend to maintain it.

**Limitations:**

yes

**Paper Formatting Concerns:**

The page count, margins, font size, anonymity guidelines, and checklist placement appear to follow the NeurIPS 2025 style guidelines.

**Quality:**

3

**Strengths And Weaknesses:**

Strengths:
- The paper clearly and concisely presents the problem background, objectives, and proposed method, making it very easy to follow. Differences from prior work are explicitly discussed, and the novelty of the approach is well-articulated.
- Training spatial and temporal information with a single encoder is intuitive and convincing. The paper explains the rationale behind each design choice, and the method achieves strong results across diverse image and video benchmarks, demonstrating its effectiveness. In addition, large-scale scaling experiments (data volume, model capacity, input resolution) show a consistent, monotonic performance gain.
- In the verification, the effectiveness of each feature of the proposed method, such as U-RoPE, was confirmed through various ablations, enhancing the reliability of this study.

Weaknesses:
- The method is claimed to learn object-level representations for images and event-level representations for videos. Although ablations show that combining both improves performance, further analysis is needed to verify that the learned features (or clusters) truly capture these properties. In particular, it should be analyzed what kind of feature representations can be obtained for long-term and short-term events.
- As the authors also point out, since clusters are determined offline, biases from initial feature extraction remain, posing challenges related to domain shift.

---

> ### Author Rebuttal · Authors · 2025-07-30
>
> >**W1: The method is claimed to learn object-level representations for images and event-level representations for videos. Although ablations show that combining both improves performance, further analysis is needed to verify that the learned features (or clusters) truly capture these properties. In particular, it should be analyzed what kind of feature representations can be obtained for long-term and short-term events.**
>
> **A1:** Thank you for pointing out the need for deeper analysis of the learned object-level and event-level representations. We acknowledge that interpreting feature behavior across different temporal scales is important, and we have added detailed visualization and clustering analyses to better understand how our model captures both short-term and long-term event structures.
>
> Our current evaluation primarily focuses on quantitative results (e.g., performance on K400 and SSV2) and qualitative clustering visualizations (Fig. 6 and Fig. 7), which demonstrate the model’s effectiveness in unifying image and video representations. Our goal is to learn representations at multiple semantic granularities: for example, from single-frame inputs to 16-frame clips, we expect the model to capture both global semantics and fine-grained object-level details.
>
> To analyze this, we examined cluster visualizations. We observed that object-level clusters often capture single-object semantics (e.g., "cake", "wedding dress") or simple multi-object scenes (e.g., "a child with a backpack"), while event-level clusters correspond to action or event semantics (e.g., "mixing cement"). This suggests the model is indeed separating visual concepts across semantic levels. We will include representative cluster visualizations in the final version to further support this analysis.
>
> While our current analysis offers initial insights, it does not yet provide fine-grained verification of the temporal specificity of event representations, particularly in distinguishing long-term versus short-term dynamics. Our framework is designed with the intent that long-range segments encode high-level, holistic event semantics, whereas short-term segments capture finer-grained procedural steps or sub-actions. To facilitate this, we adopt an offline clustering strategy with a high-capacity vocabulary (1 million clusters), enabling the generation of detailed pseudo-labels that promote segment-level semantic learning. We consider this an important direction for future research and will provide a more in-depth discussion on temporal decomposition analysis in the final version of the paper.
>
> >**W2: As the authors also point out, since clusters are determined offline, biases from initial feature extraction remain, posing challenges related to domain shift.**
>
> **A2:** Thank you for your constructive comment, this indeed touches on a key limitation of our method. As you pointed out, the clustering process is performed offline based on initial features, meaning the model cannot dynamically update representations during training. This can potentially introduce bias from the initial features and affect generalization, especially under domain shifts.
> However, we would like to clarify that generating pseudo-labels via clustering introduces only a weak and indirect dependency on the initial model, in contrast to direct distillation from a teacher model. This mitigates the influence of the initial features to some extent.
> Our choice of an offline clustering strategy was a trade-off between training stability and computational efficiency. Compared to iterative clustering methods like DeepCluster [1], our approach maintains scalability and avoids the instability that may arise from end-to-end joint optimization on large-scale datasets.
> We recognize that the risk of bias still exists. To address this, we carefully selected the initial model based on findings from the MetaCLIP [2] paper, opting for CLIP, which has been shown to be among the least biased available models.
> Incorporating iterative or dynamically updated clustering could further enhance semantic modeling and cross-domain robustness. This is indeed a key direction for our future work, and we will make this limitation and its potential solutions clearer in the revised paper.
>
> > **Q1: I would like the authors to address the weaknesses I pointed out.**
>
> **A1:** Thank you again for your thoughtful feedback. We have carefully addressed each of the concerns you raised, including verifying the object-level and event-level representations, analyzing short-term and long-term temporal structures, and mitigating biases introduced by offline clustering. We hope our detailed responses and new experimental results help resolve the issues and demonstrate the strengths and robustness of our approach. We would be grateful if you could consider updating your rating and confidence accordingly.
>
>
> > **Q2: The paper states that a 1:1 ratio between images and videos was maintained during training. Does this ratio refer to the number of images versus the number of video frames (e.g., 2 K videos × 16 frames = 32 K frames, which would not match the number of images), or is it defined by the number of tokens after aggregation with VS²? Please clarify this design choice and the reasoning behind it.**
>
> **A2:** Thank you for the detailed question. We follow the first interpretation you mentioned, the 1:1 ratio refers to the number of images versus the number of video frames. For example, 2K videos × 1 clip (e.g., 16 frames) are treated as 2K video samples, matched with 2K image samples.
> We conducted preliminary ablation studies to compare different image-to-video data ratios. Our design prioritizes improving performance on video tasks while maintaining competitive image performance, aiming to enhance video modeling without sacrificing image understanding, as compared to MLCD. The table below shows the results on UniViT-S with varying data ratios.:
>
> **Table 1:** Ablation results on different image-to-video sample ratios, showing that a balanced 1:1 ratio provides a good trade-off between video and image performance.
> | Data (I:V) | SSV2 | K400 | ImageNet |
> | ----------- | ---- | ---- | -------- |
> | 1:3         | 16.4 | 64.2 | 81.3     |
> | 1:1         | 16.8 | 67.6 | 80.4     |
> | 3:1         | 16.9 | 67.8 | 79.8     |
>
> These results suggest that using a balanced 1:1 image-to-video sample ratio yields the most favorable trade-off between video and image performance. While increasing the video ratio (1:3) slightly improves SSV2 performance, it leads to a drop in image accuracy. Conversely, increasing the image ratio (3:1) compromises video modeling. This supports our design choice of maintaining a balanced data composition to ensure strong video understanding without significantly degrading image-level representation learning.
>
> References:
>
> [1] *Deep Clustering for Unsupervised Learning of Visual Features*
>
> [2] *Demystifying CLIP Data*

---

> > ### Author Response · Authors · 2025-08-04
> >
> > Dear Reviewer LTUC,
> >
> > We sincerely appreciate the time and effort you have invested in reviewing our submission. Your insightful feedback has been invaluable to us, and we have diligently worked to address all the concerns you raised in our rebuttal. As the author-reviewer discussion phase is drawing to a close, we would like to confirm whether our responses have effectively addressed your concerns. We are more than happy to provide any further details or explanations. Thank you once again for your thoughtful review and consideration.
> >
> > Best regards,
> >
> > The Authors

---

> > > ### Comment · Reviewer_LTUC · 2025-08-07
> > > **Official Comment by Reviewer LTUC**
> > >
> > > Thank you for your thorough response. My remaining concerns have been largely resolved. In particular, A2 clarified that the current data ratio offers the best balance, which I found convincing. Since I have already given an Accept rating, I intend to maintain it.

---

> > > > ### Author Response · Authors · 2025-08-07
> > > >
> > > > Thank you very much for your kind response and for taking the time to engage with our work so thoughtfully. We truly appreciate your careful reading and constructive comments throughout the review process. We’re very pleased to hear that your remaining concerns have been addressed, and your recognition is genuinely encouraging for our team.
> > > >
> > > > If you have no further concerns, would you consider improving your confidence or rating score for our work?  We would be deeply grateful if you are willing to update your score or confidence rating. We are more than happy to provide any further details or explanations. Thank you again for your thoughtful review and consideration.

---

### Note · Authors · 2025-08-13

Dear ACs and SAC:

We sincerely thank the AC and SAC for guidance and the reviewers for their constructive and thoughtful feedback throughout the review process.

**Contribution:**

**(i)** UniViT introduces a unified single-encoder architecture that explicitly models both **spatial details (object-level)** and **temporal dynamics (event-level)** through clustering and discrimination, event modeling across varied temporal spans, and structurally constrained contrastive objectives. **(ii)** The framework integrates U-RoPE to disentangle spatial–temporal positional information and  VS² to adaptively handle varying frame lengths, enabling position-invariant and structurally consistent spatiotemporal representations. **(iii)** This structured unified semantic learning mechanism consistently improves performance on both **image and video benchmarks** without trade-offs, extends effectively to **MLLMs**, and is further supported by extensive visualization and analytical experiments validating its effectiveness.

**Reviewer Responses:**

- **LTUC:** Found problem definition clear and well-motivated, design intuitive and convincing; clustering concerns fully resolved; inclined to `accept`.

- **FTq5:** Recognized solution effectiveness in unified modeling; open-sourcing plan reaffirmed; inclined to `weakly accept`.

- **mwRn:** Recognized the novelty and reasonable design of U-RoPE, with the main concern focusing on overall novelty. We addressed this with detailed clarifications on methodological distinctions, evidence of consistent cross-modal performance gains, and supplementary SSV2 experiments demonstrating substantial improvements in temporal modeling. Although the reviewer indicated a `score increase` during discussions, this was not reflected in the final rating; we believe the current assessment may still underestimate the strength and originality of our contributions.

- **Sxna:** Praised novel clustering strategy and evaluation coverage; setup concerns resolved with supplemental experiments; raised score to `weak accept`.

- **Wzoa:** Highly valued problem importance, motivation, and technical completeness; score remained `accept`.

We sincerely appreciate the thoughtful time and effort that all reviewers, the AC and SAC have dedicated to the review and discussion process. We believe the concerns raised have been alleviated, with resulting score improvements, and we will incorporate the recommendations into the final version.

Best regards,

The Authors

---

### Decision · Program_Chairs · 2025-09-17

**Decision:**

Accept (poster)

**Comment:**

### Summary

This paper introduces UniViT, a unified vision encoder for both images and videos. The authors claim that by performing object-level clustering for images and event-level clustering for videos, their model can effectively learn both image and video representation within a single architecture. The paper also introduces two novel components: Unified Rotary Position Embedding (U-RoPE) to better handle spatiotemporal positional information and Variable Spatiotemporal Streams (VSS) to accommodate video inputs of varying lengths. The authors present extensive experimental results showing that UniViT achieves state-of-the-art performance on a variety of tasks.

### Strengths
- The paper tackles the important and challenging problem of creating a single, unified model for both image and video representations.
- Novel and solid method. The proposed method, which combines clustering-based self-supervision on both images and videos with contrastive learning, is technically solid. The introduction of new positional embedding U-RoPE and VSS for are necessary additions to address specific challenges in unifying these two modalities.
- The authors have conducted extensive experiments across a wide range of benchmarks for images, videos, and multimodal tasks. The proposed method demonstrates state-of-the-art or highly competitive performance across most of the evaluated tasks.

### Weaknesses
- Novelty Concerns: mwRn has concerns about the technical novelty, suggesting it is a straightforward extension of a previous method, MLCD.
[1] Multi-label Cluster Discrimination for Visual Representation Learning.

- Clarity and Presentation: Reviewer mwRn also pointed out some clarity issues, with terms like "object" and "event" not being precisely defined.
- Reliance on Offline Clustering: The method's dependence on an offline clustering step using a pretrained model was noted as a potential weakness by reviewer FTq5, as it could introduce biases from the initial model.
- Initial Discrepancies in Baseline Results: Reviewer Sxna raised valid concerns about discrepancies in the reported performance of some baseline models.

### Reasons for Recommendation
I recommend this paper for acceptance. The problem of unified image and video understanding is of high importance to the community, and this paper presents a solid and effective solution. The extensive and strong empirical results across a diverse set of tasks are a significant contribution. The authors have also been very responsive and thorough in their rebuttals, addressing most of the reviewers' concerns with additional experiments and clarifications.
Despite the concerns regarding novelty, the strengths of the paper outweigh the weaknesses. While the core idea builds upon existing work, the specific adaptation for the video domain and the proposed architectural modifications (U-RoPE and VSS) provide sufficient novelty. Majority of the reviewers are also biased towards an Accept.

### Summary of Rebuttal and Discussion
- Reviewer mwRn's main concerns were about the novelty of the work in relation to MLCD and the clarity of the writing. The authors provided a detailed breakdown of the differences between UniViT and MLCD in terms of input, architecture, and methodology. They also clarified the definitions of "object-level" and "event-level" clustering and provided more details on their experimental setup. Despite the extensive discussion, this reviewer remained concerned about the overall novelty, though they did raise their score , acknowledging the authors' efforts.
- Reviewer Sxna questioned the discrepancies in baseline performance and the temporal modeling capabilities on datasets like Something-Something V2. The authors clarified that the discrepancies were due to different evaluation protocols (e.g., 50-shot probing vs. full fine-tuning) and provided additional experimental results, including full fine-tuning comparisons, to address these concerns. These clarifications were convincing, and the reviewer raised their score to "Borderline accept."
- Reviewer FTq5 pointed out the reliance on offline clustering and initial concerns about reproducibility. The authors addressed the offline clustering by explaining the trade-offs and committed to releasing their code and models, which satisfied the reviewer.
- Reviewers LTUC and Wzoa were generally positive from the start. Reviewer LTUC's concerns about the verification of learned features and the data ratio were addressed in the rebuttal. Reviewer Wzoa had questions about missing baselines and implementation details, which the authors also addressed by adding more experimental comparisons and clarifying their methodology.

This paper received mixed ratings. But AC believes the authors provide a strong rebuttal. The fact that they were able to convincingly address the concerns of most reviewers, leading to score increases, speaks to the solidity of their work. While the novelty debate with reviewer mwRn is a valid point of discussion,  AC and other reviewers think that this work has sufficient novelty compared to MLCD. The overall contribution and the quality of the empirical results are strong enough to warrant acceptance.